# The Genus *Chiliadenus*: A Comprehensive Review of Taxonomic Aspects, Traditional Uses, Phytochemistry and Pharmacological Activities

**DOI:** 10.3390/plants14020205

**Published:** 2025-01-13

**Authors:** Lucinda Villaescusa Castillo, Francisco Zaragozá García, Cristina Zaragozá Arnáez

**Affiliations:** Pharmacology Unit, Biomedical Sciences Department, Faculty of Pharmacy, University of Alcalá, 28871 Alcalá de Henares, Spain; francisco.zaragoza@uah.es

**Keywords:** *Chiliadenus*, *Jasonia*, folk medicine, bioactive compounds, biological activities

## Abstract

The genus *Chiliadenus* (Asteraceae) has been traditionally used in Mediterranean medicine for its anti-inflammatory, antioxidant, and antimicrobial properties. However, scientific research on this genus remains limited, highlighting the need for a comprehensive review of its chemical composition and pharmacological characteristics. This review compiles existing knowledge on *Chiliadenus* species, focusing on their secondary metabolites, such as flavonoids, terpenes, and essential oils, as well as associated biological activities. The findings show that the traditional therapeutic properties of *Chiliadenus* are well supported by reported pharmacological activities in previous studies, emphasizing the potential of this genus for the development of new therapeutic agents. However, the lack of comparative studies among *Chiliadenus* species and the scarcity of in vivo studies and clinical trials hinder the full realization of its therapeutic potential. Specifically, comparative studies could be key to identifying species with unique chemical profiles and understanding how variations in secondary metabolite composition may influence their pharmacological activities. This work highlights the urgent need to expand research in these areas to validate the pharmacological properties of *Chiliadenus* species for their application in modern medicine.

## 1. Introduction

Natural products have persisted as an inexhaustible source of bioactive molecules with a structural diversity that far exceeds that of conventional synthetic compounds. Throughout history, these compounds have played an essential role in the discovery of new and interesting drugs, as their complex chemical structures—often unsuspected—have proven to be highly effective in the treatment of numerous diseases. In fact, it is estimated that more than 50% of the medicines approved in recent decades are directly or indirectly derived from natural products, underlining the importance of exploring the world’s flora to identify new chemical entities with therapeutic potential [1].

The ability of natural products to interact selectively with specific proteins has been one of the key factors that make them so valuable in drug discovery. In contrast to synthetic compounds, natural products tend to exhibit greater molecular rigidity and structural complexity, resulting in more specific and potent biological interactions. In addition, the richness in carbon and oxygen atoms of these molecules, together with a lower content of nitrogen atoms and halogens, suggests a favourable pharmacokinetic profile, with fewer adverse effects and better bioavailability. In this context, ethnopharmacology, as a discipline that studies the traditional use of medicinal plants, has established itself as an indispensable tool to guide the search for new drugs. The correlation between traditional uses and modern pharmacological activities is remarkable; one study found that 80% of plant-derived compounds used therapeutically have therapeutic indications consistent with their ethnopharmacological use. This finding highlights the validity of traditional knowledge as a starting point for pharmacological research and the importance of systematically documenting and analyzing ancestral medicinal practices [1].

The *Asteraceae* family, with more than 25,000 species distributed globally, stands out as one of the most prolific botanical families in terms of the production of bioactive compounds. Many of the species are widely used in traditional medicine and have been an important source of bioactive compounds that have led to the development of pharmaceuticals. One of the most notable examples is artemisinin, a sesquiterpene lactone extracted from *Artemisia annua*, which revolutionized the treatment of malaria. In addition, other species have been shown to possess a wide range of therapeutic properties, including anti-inflammatory, antioxidant, antiviral, antiprotozoal, hepatoprotective, wound-healing and antitumor properties, among many others [2]. Today, the *Asteraceae* family remains an invaluable source of chemical biodiversity, with great potential for the advancement of new industries, supported both for the traditional employment of its species or for scientific studies confirming their diverse therapeutic properties.

The *Asteraceae* family is the biggest family of blooming plants. It involves around 1600 genera as well as about 25,000 species distributed worldwide (https://wfoplantlist.org/ accessed on 18 December 2024). Within this family, *Chiliadenus* Cass (tribe Inuleae) is a small genus consisting of 10 species, many of which have been widely used in traditional medicine for different therapeutic purposes in countries of the Mediterranean area. These species, mainly distributed in arid and semi-arid areas, have been used to treat a variety of conditions, including inflammatory conditions, infections and digestive disorders. However, despite their therapeutic potential, scientific knowledge about the chemical and pharmacological properties of *Chiliadenus* species is still limited. Work on the genus *Chiliadenus* has documented the presence of several classes of bioactive compounds, such as flavonoids, terpenes and essential oils, which may be responsible for its therapeutic effects.

From a taxonomic point of view, the species of this genus used to be included in *Jasonia* Cass and/or *Varthemia* DC, although Brullo (1979) transferred them to the genus *Chiliadenus*, supported by their structure, but not all authors have considered these synonymies in their works.

In 2020, Shaaban et al. published a review of the chemical components isolated from the different species of the genus *Chiliadenus* [3]; however, this review does not take into account the synonymies between the genera *Chiliadenus* and *Jasonia*, which could lead to an omission of relevant data on the compounds and their biological activity, and consequently to an underestimation of the pharmacological potential of these plants.

Despite the recognized chemical and biological diversity of the genus *Chiliadenus*, there is a notable lack of comparative studies that systematically assess the differences and similarities between the species that comprise it. This lack of research has limited the comprehensive understanding of the genus, as most studies have focused on individual species, without establishing comparisons that could reveal key biological or chemical patterns. Such a comparative approach could be crucial in identifying species with unique chemical profiles, and to understand how variations in secondary metabolite composition may influence their respective pharmacological activities. Furthermore, despite the therapeutic potential suggested by in vitro studies and traditional use of *Chiliadenus*, there is a worrying paucity of work that has progressed to in vivo models or clinical trials. This gap in clinical research not only limits the scientific validation of ethnopharmacological uses, but also hinders the potential development of new drugs based on the bioactive compounds of this genus. Clinical screening is essential to determine the efficacy, safety and appropriate dosage of *Chiliadenus* extracts and isolated compounds, which are essential steps before any application in modern medicine can be considered.

Given the increasing importance of ethnopharmacological studies in the identification of new drugs and the underestimated potential of *Chiliadenus* species, this work aims to comprehensively review the ethnopharmacological data, chemical composition and biological activity of extracts and compounds isolated from the different species of this genus. This review aims to provide a broader and deeper understanding of these medicinal plants, offering a solid basis for future research that may lead to the discovery of new therapeutic agents derived from *Chiliadenus*.

## 2. Methodology

To analyze the traditional uses, phytochemical aspects, and biological activity of plants from the *Chiliadenus* genus, a comprehensive bibliographic review was conducted using various online databases, including PubMed, Web of Science, Scopus, ScienceDirect, SciFinder, Wiley Online Library, ACS, SpringerLink, and Google Scholar. The search was designed to identify all available studies related to *Chiliadenus* and *Jasonia* genera, as well as their synonymies.

Search Terms and Descriptors

The following primary search terms were used to ensure comprehensive coverage of the topic: *Chiliadenus*, *Jasonia*. These terms were used in combination to locate all publications related to these genera, including studies on species that are synonymously classified under both genera. Furthermore, additional search terms were used to ensure that relevant studies on synonymously related species were not omitted. These include: *Inula*, *Erigeron* and specific species: *Inula saxatilis* and *Erigeron glutinosus* (known synonyms of *Chiliadenus* and *Jasonia* species).

To include information on the taxonomic classification and potential synonymies, searches were also performed using alternative taxonomic names, such as *Varthemia montana* and *Varthemia iphionoides* for *Chiliadenus montanus* and *Chiliadenus iphionoides*, respectively, based on their former classification.

Additionally, other search terms related to the chemical composition and biological activity of the plants were included: Traditional uses, Chemical composition and Pharmacological activity.

The following inclusion criteria were applied to select relevant articles:-Studies discussing the taxonomic classification of *Chiliadenus* and its synonymies with *Jasonia*.-Publications that provide detailed information on the chemical composition, phytochemical properties, and bioactive compounds isolated from *Chiliadenus* and related species.-Articles examining the pharmacological activities of *Chiliadenus* species, including in vitro studies, clinical studies, or ethnopharmacological reviews.-Studies focusing on traditional uses of *Chiliadenus* species in medicinal practices.

Exclusion criteria included:-Articles not related to the species of *Chiliadenus*, *Jasonia*, or their known synonyms.-Studies that focused on non-ethnopharmacological uses or did not provide sufficient data on chemical or pharmacological properties.

Database Search and Article Selection

To ensure a comprehensive review of the literature, no time limits were set for the search years, allowing the inclusion of both recent and older studies. After applying the inclusion and exclusion criteria, articles that provided sufficient data on the taxonomy, chemical composition, traditional uses, and pharmacological activity of the species were selected.

To guarantee the accuracy of the taxonomic classification and synonymies, global databases, such as the World Flora Online Plant List (WFO) and the Plants of the World Online (POWO), maintained by the Royal Botanic Gardens, Kew, were consulted. These resources provided up-to-date and authoritative information on the plant species discussed, including the correct taxonomic names and synonymies.

## 3. Taxonomic Considerations on the Genus *Chiliadenus* and Its Relationship to the Genus *Jasonia*

The genus *Jasonia* is included in the tribe Inuleae, a complex group of species belonging to the family Asteraceae. French botanist Alexandre de Cassini first described this genus. About 20 species were included, including *Jasonia longifolia* Cass., *Jasonia radiata* Cass. and *Jasonia tuberosa* (L.) DC. *Jasonia glutinosa* has been classified in the genus *Chiliadenus*, as *Chiliadenus glutinosus* (L.) Fourr. with the following synonymies: *Jasonia glutinosa* var. *glutinosa*, *Jasonia glutinosa* (L.) DC., *Chiliadenus saxatilis* (Lam.) Brullo, *Jasonia saxatilis* (Lam.) Guss., *Chiliadenus camphoratus* Cass. and *Inula saxatilis* Lam. [Global Compositae Checklist (GCC, www.compositae.org/checklist, accessed on 19 August 2024)]. Alexandre de Cassini later described the genus *Chiliadenus* for the first time to replace that of its former genus *Myriadenus* Cass. Cassini includes the genus *Chiliadenus* within the tribe Inuleae, order Asterales, family Asteraceae, and this is integrated by ten species from selected regions of the Mediterranean zone [4]. (https://www.catalogueoflife.org/data/taxon/3MZM, accessed on 19 August 2024) (https://www.worldplants.de/world-plants-complete-list/complete-plant-list/?name=Chiliadenus, accessed on 19 August 2024).

Table 1 shows the species belonging to the genus *Chiliadenus*, as well as the accepted synonymies.

Cassini initially described the genera *Jasonia* and *Chiliadenus* separately, but later taxonomic revisions led to the reclassification of some species. In many cases, species originally described within the genus *Jasonia* were transferred to the genus *Chiliadenus.*

In 1836, De Candolle grouped four sections under the genus *Jasonia*, two of which correspond to *Chiliadenus* (with *Jasonia glutinosa*) and *Eujasonia* (with *Jasonia tuberosa* and *Jasonia sicula*) [5]. Later, in 1979, Brullo, in his work on the genus *Chiliadenus*, recognized nine species, excluding *Jasonia tuberosa*. Brullo argued that these species should not be included in the genus *Jasonia* Cass., but rather should be assigned to the genus *Chiliadenus* Cass., which differs significantly from the former by the absence of ligulate flowers and the presence of achenes covered in glands and hairs. Upon reviewing numerous herbarium specimens of *Jasonia glutinosa* (L.) DC. from various locations, including the Iberian Peninsula, Lampedusa, Malta, the Balearic Islands, Morocco, and France, Brullo found notable differences between the Franco-Iberian and Maltese plants, particularly in the stem shape, hair distribution, capitula arrangement on the scape, leaf structure, achenes, and involucre bracts. These distinctions confirmed that they were two separate species, revealing a confusing situation concerning their nomenclature. In addition to *Chiliadenus saxatilis*, with a distribution area from the south of France, Mallorca, Spain and Morocco, it considers as new species the populations of Malta (*Chiliadenus bocconei*) and Lampedusa (*Chiliadenus lopadusanus*). It also includes *Chiliadenus candicans* (Delile) Brullo from the northern coast of Egypt, *Chiliadenus montanus* (Vahl) Brullo from the Sinai Peninsula, and *Chiliadenus iphionoides* (Boiss. and Blanche) Brullo from Egypt, Palestine and Lebanon, which were considered as a separate genus, *Varthemia* (*Varthemia montana* (Vahl) Boiss., *Varthemia candicans* (Delile) Boiss. and *Varthemia iphionoides* Boiss. and Blanche in Boiss.

From these investigations, Brullo concludes that the genus *Jasonia* is monotypic and is depicted by *J. tuberosa* (L.) DC. (*Erigeron tuberosum* L.) and, on the other hand, the genus *Chiliadenus* includes these other types: *Chiliadenus bocconei* Brullo sp. nov. (*Orsina camphorata* Bertol. pp.), *Chiliadenus saxatilis* (Lam.) Brullo comb. nov. (*Inula saxatilis* Lam.), *Chiliadenus candicans* (Delile) Brullo comb. nov. (*Varthemia candicans* [Delile] Boiss.), *Chiliadenus lopadusanus* Brullo sp. nov., *Chiliadenus montanus* (Vahl) Brullo comb. nov. (*Varthemia montana* [Vahl] Boiss.), *Chiliadenus iphionoides* (Boiss. and Blanche) Brullo comb. nov. (*Varthemia iphionoides* Boiss. and Blanche), *Chiliadenus hesperius* (Maire and Wilczek) Brullo comb. nov. (*Jasonia hesperia* Maire and Wilczek), *Chiliadenus sericeus* (Batt. and Trabut) Brullo comb. nov. (*Jasonia sericea* Batt. and Trabut), *Chiliadenus rupestris* (Pomel) Brullo comb. nov. (*Jasonia rupestris* Pomel) (Table 1) [6].

According to the criterion proposed by Brullo, *J. glutinosa* and *J. tuberosa* belong to different genera and, in this sense, with respect to the former, he considers it more appropriate to include it within the genus *Chiliadenus Cass*, under the name *Chiliadenus saxatilis* (Lam.) Brullo, since its capitula have only flosculose flowers, as well as achenes with a setae pappus with a single row of hairs, characteristics that he considers of great taxonomic importance and justify its inclusion in a genus different from *Jasonia* [7].

However, in 2004, Pardo de Santayana proposed the inclusion of this plant in the genus *Jasonia*, since he considers that the characteristics indicated by Brullo (1979) for the division of these two genera are not significant [8]. Scientific names of plants, based on binomial nomenclature, are often revised and adjusted as additional research is conducted and evolutionary relationships among species are better understood. In 2018, Bengtson et al. conducted a molecular phylogenetic study, as well as a biogeographical analysis of the different species composing the genus *Chiliadenus*, which led to the confirmation of the monophyletic character of this genus and of its divergence from the genus *Dittrichia*, dispersing across the Mediterranean to its current distribution [9]. In view of this confirmation, this work aims to carry out an exhaustive review of the botanical characteristics, geographic distribution, traditional uses, chemical composition and biological activities of the different species that make up the genus *Chiliadenus.*

## 4. Description of the Species of the Genus *Chiliadenus*

*Chiliadenus* Cass. is a small genus belonging to the tribe Inuleae–Inulineae (*Asteraceae*) that includes ten species distributed mainly along the southern edge of the Mediterranean Sea (Figure 1). Most of its species grow in rocky places and semi-arid areas [6,9]. 

The genus *Chiliadenus* is composed of perennial woody plants or shrubs with yellow flowers and discoid flower heads and these are characterized by a pappus with double rows of hairs (Figure 2) [10]. Phytochemical research of the genus has led to the identification and isolation of numerous bioactive compounds belonging to various chemical classes. The botanical characteristics, geographical distribution, traditional uses, chemical composition (mainly focused on the non-volatile components of the extracts) and biological activities of the different species of this genus are described below. Table 2 summarizes the main pharmacological activities described for the different species of the genus *Chiliadenus*.

### 4.1. Chiliadenus glutinosus

*C. glutinosus* (L.) Fourr. [*J. glutinosa* (L.) DC, *J. saxatilis* (Lam) Guss, *C. saxatilis* (Lam.) Brullo, *Erigeron glutinosus* L., *Inula saxatilis* Lam] is a species popularly known as “rock tea” or “Aragon tea”. It is the only species of the genus *Chiliadenus* that grows in Spain, distributed in the island of Mallorca, in Levante and extending to north Morocco and the south of France (Figure 1A) [6,8,11]. It lives exclusively in fissures and landings of vertical calcareous rocks or in horizontal crevices, generally below 1300 m in altitude [8]. It is a perennial plant, about 60 cm high, with plenty of trichomes, several of which are short glandular and others long non-secretory, with a branched or simple stem and a slender rhizome with narrow and long roots that hide among rocks. Its leaves are alternate, lanceolate, small, sharp, erect, entire or rarely toothed, and are covered with glandular hairs. The flower heads are grouped in cymose or corymbose inflorescences and occasionally isolated. The involucre has bracts arranged in several rows. In each flower head there are numerous yellow florets (Figure 2A). The pappus shows two lines of hairs. Flowering occurs from May to October, mainly between July and August [12,13,14]. It should be harvested in these months when the plant is flowering; the flowering aerial part is easily collected by breaking the stems, leaving the base intact. It is quite abundant and, despite intensive harvesting, natural populations are not endangered [15].

#### 4.1.1. Traditional Uses

The first information on the use of this plant dates back to 1867 and, at present, it is one of the most popular and widely used medicinal species in the Iberian Peninsula. It is used in infusion and has a camphorated aroma and a slightly bitter taste [8]. In the places where it grows, *C. glutinosus* has a known and widespread reputation for its broad variety of medicinal uses, especially in Spain [8,12]. It is used mainly to treat digestive pathologies (stomach pain, dyspepsia, diarrhoea, gastric ulcers, vomiting, etc,) and respiratory (sore throat, asthmatic processes and respiratory infections), for the treatment of circulatory disorders (arterial hypertension, varicose veins), genitourinary disorders(as a diuretic and to treat pain due to kidney stones, etc.), as well as in musculoskeletal disorders (rheumatic diseases and joint pain, bone, etc.) and nervous system (as a tranquilizer and analgesic). In addition to these internal uses, it is also used topically, in the form of infusion or decoction in alcohol, in the treatment of skin disorders, for washing and disinfecting injuries, and in the form of a poultice to be applied to wounds and burns [8,16].

This plant has a long tradition of use in Catalonia (Spain) to prepare a liqueur called “ratafía”, which is made with brandy and other medicinal plants, together with lemon peel and spices, such as nutmeg, cloves and mint. It is consumed mainly as a digestive after meals to relieve digestive disorders and other minor problems, and as an aperitif to stimulate appetite [8].

The results of a study conducted in Navarre (Spain) on ethnobotanical knowledge of the region’s medicinal plants show that *C. glutinosus* is one of the species with the highest number of citations and a long tradition of use [17]. Subsequently, the same research group conducted an ethnopharmacological study on the medicinal plants used in Navarre for the treatment of neurological and mental disorders. The authors highlight the infusion of the aerial part of this species, for which the most reported applications were “to clear the mind” and in “headache”. This study underlines the importance of traditional knowledge of the use of medicinal plants and provides a basis for future scientific research that could validate and explain the mechanisms of action of this species in the treatment of neurological and mental disorders [8,18].

These effects on the central nervous system could be related, in part, to the antioxidant action demonstrated for this species. Antioxidant properties can protect cells from damage caused by free radicals, which are involved in oxidative stress, which, in turn, is associated with various neurological and mental disorders, as well as headaches. The antioxidant action may help reduce inflammation and protect the nervous system, improve brain function and relieve pain. On the other hand, inflammation and oxidative stress are associated with the aging progress as a causative factor in neurodegeneration among other pathologies [18].

Alarcón et al. (2015) conducted a study whose objective was to contribute to the understanding of the local and traditional use of plants in the Basque Country (Spain), a region that preserves a rich heritage in the employment of medicinal herbs. One of the most frequently reported species in this study is *C. glutinosus*. An infusion of this species is the most popular tea and has great importance in the local phytotherapy [19].

#### 4.1.2. Phytochemistry

*C. glutinosus* has been studied by researchers all over the world, but most research has been conducted in Spain. Phytochemical studies have conducted on the aerial part of the plant, since it is that part traditionally used in folk medicine. Generally, the plant is extracted once dried, except when investigating the composition of the essential oil, in which case the fresh plant is used and extracted by hydro-distillation. The extraction of the non-volatile components of the plant is carried out using different methods, generally maceration at room temperature, and the solvents used are varied depending on the type of components to be extracted. Methanol, ethanol, hydroalcoholic mixtures, acetone or apolar solvents, such as benzene or dichloromethane, have been used. A large number of different chemical compounds have been identified in this species to date, mainly belonging to the group of phenolic compounds and terpenes (monoterpenes and sesquiterpenes) (Table 2).

For consistency with the existing literature and for ease of understanding, we will use the name *J. glutinosa* if studies have been published using that name. By using the name under which the articles have been published, we seek to ensure greater clarity and precision in the discussion of the data and findings in the bibliography.

**Table 2 plants-14-00205-t002:** Main chemical components identified in *J. glutinosa* (*C. glutinosus*).

Compound	Reference
Flavonoids	
Patuletin-7-O-β-D-glucopyranoside (**1**)	[20]
Patuletin-3-O-β-D-glucopyranoside (**2**)	[20]
Quercetin-3-O-β-D-glucopyranoside (**3**)	[20]
Quercetin-3-O-β-D-galactopyranoside (**4**)	[20]
Quercetin-7-O-β-D-glucopyranoside (**5**)	[20]
Kaempferol-3-O-β-D-glucuronopyranoside (**6**)	[21]
Quercetin-3-O-β-D-glucuronopyranoside (**7**)	[21]
Kaempferol-3-O-β-D-glucuronopyranoside-6″-methyl ester (**8**)	[21]
Terpenes	
(−)-[11R]-4α-hydroxy-eudesm-11,12-diol (kudtdiol) (**9**)	[22]
(−)-[11R]-4α,14-epoxyeudesm-11,12-diol (α-epoxy kudtdiol) (**10**)	[23]
(−)-[11R]-eudesm-4(14)-en-5β,11,12-triol (5-epi-kudtriol) (**11**)	[23]
(+)-[11R]-eudesm-4(14)-en-5α,11,12-triol (kudtriol) (**12**)	[23]
5β, ll,12-trihydroxy-iphionan-4-one (lucinone) (**13**)	[24]
(2R)-2-methyl-2-(4-oxopentyl)-5-(1-methyl-1,2-dihydroxyethyl) cyclohexanone (glutinone) (**14**)	[24]
(11 R)-eudesm -4-en-11,12-diol (**15**)	[25]
(11 R)-eudesmane-5α,11,12-triol (**16**)	[25]

The total polyphenol content is high, although it should be noted that there are variations, as it is influenced by the solvent used for extraction, being more abundant in hydroalcoholic extracts.

In 1995, our research group described the presence of five flavonoids from the CH_3_OH/H_2_O (60/40) extract of the aerial part of *J. glutinosa*. Their structures were established on the basis of the UV (ultraviolet) spectral data, TLC (Thin Layer Chromatography), and ^1^H, ^13^C NMR (Nuclear Magnetic Resonance). The flavonoids were patuletin-7-O-β-D-glucopyranoside (**1**), patuletin-3-O-β-D-glucopyranoside (**2**), quercetin-3-O-β-D-glucopyranoside (**3**), quercetin-3-O-β-D-galactopyranoside (**4**) and quercetin-7-O-monoglucoside (**5**) (Figure 3). The presence of methoxylated flavonoids allowed us to raise some chemotaxonomic hypotheses, since these compounds had not been described thus far in the genus *Jasonia* [20]. Likewise, in the butanolic extract of the aerial part, the flavanol glucuronides kaempferol-3-O-β-D-glucuronopyranoside (**6**), quercetin-3-O-β-D-glucuronopyranoside (**7**) and kaempferol-3-O-β-D-glucuronopyranoside-6”-methyl ester (**8**) were identified (Figure 3) [21].

Other products present in this plant have also been of interest. The sesquiterpene compounds of the eudesmanolide family have attracted the attention of numerous researchers because of their diverse range of particularly significant biological activities. In 1978, Pascual Teresa et al. isolated and identified (−)-[11R]-4α-hydroxy-eudesm-11,12-diol (kudtdiol) (9), a bicyclic sesquiterpene alcohol [22], and more identifications would follow. In 1980, three sesquiterpene alcohols were isolated. Their structures were elucidated by spectroscopic methods and chemical correlations as (−)-[11R]-4α,14-epoxyeudesm-11,12-diol (10), (−)-[11R]-eudesm-4(14)-en-5β,11,12-triol **(11)** and (+)-[11R]-eudesm-4(14)-en-5α,11,12-triol **(12)** and they are called α-epoxy kudtdiol, 5-epi-kudtriol and kudtriol, respectively (Figure 4) [23].

In 2000, kudtriol and 5-epi-kudtriol, as well as their epimers at position 11, were synthesized in the laboratory from 1-α-santonin 4 using Sharpless asymmetric dihydroxylation as the key reaction. By comparing the spectral data of the natural triols and the synthetic samples, the absolute configuration of the natural triols could be confirmed [26].

From a CH_2_Cl_2_ extract of *J. glutinosa* our research team isolated and characterized two new eudesmanolide-type sesquiterpenes, 5β,ll,12-trihydroxy-iphionan-4-one (lucinone) (**13**) and (2R)-2-methyl-2-(4-oxopentyl)-5-(1-methyl-1,2-dihydroxyethyl) cyclohexanone (glutinone) (**14**) not identified so far in any other species (Figure 4). In this case, the alcohols are not actually alcohol but ketone derivatives. Their identification was carried out by chromatographic and spectroscopic techniques [FAB MS (Fast Atom Bombardment Mass Spectrometry), 1D and 2D NMR (One-Dimensional and two-Dimensional NMR): COSY (COrrelation SpectroscopY), HMQC (Heteronuclear Multiple Quantum Coherence) and HMBC (Heteronuclear Multiple Bond Correlation)] [24].

In 2001, Chiu et al. developed enantioselective methods for the total chemical synthesis of lucinone, which represented a significant challenge in asymmetric synthesis. This breakthrough not only confirmed the absolute configuration of these natural sesquiterpenes but also paved the way for the creation of new derivatives with potential bioactive applications [27]. That same year, Zhang et al. successfully synthesized glutinone for the first time, along with its epimers at positions C(7) and C(11), further advancing the understanding of these compounds [28].

Continuing with the study of the terpenic compounds present in this species, our research group isolated and identified two new sesquiterpene alcohol eudesmanolides from the acetone/water extract of the aerial part. This extract was fractionated by open column chromatography and, after further purification by medium pressure liquid chromatography (MPLC), the two compounds were obtained and identified by 1D and 2D NMR spectroscopic techniques [DQCOSY (Double Quantum Correlation Spectroscopy), TOCSY (Total Correlation Spectroscopy), NOESY (Nuclear Overhauser Effect Spectroscopy), HMQC and HMBC] as (11 R)-eudesm -4-en-11,12-diol (15) and (11 R)-eudesmane-5α,11,12-triol (16) (Figure 4) [25].

The chemical composition of the essential oil of the leaves of *J. glutinosa* has also been investigated by several authors. In 1996, Guillén and Ibargoitia conducted the first study on the composition of essential oils and the pentane extract derived from the leaves, using GC–MS and GC techniques. Their findings revealed that, while the leaves contain low levels of terpenic and sesquiterpenic hydrocarbons, they are abundant in their oxygenated derivatives. The main components identified in the essential oil were camphor, endo-borneol, α-terpineol, nerolidol and τ-cadinol. The principal volatile compounds recognized in the pentane extract were α-pinene, camphor, endo-borneol, γ-selinene, nerolidol, borneol formate, nonacosane and hentriacontane. The authors point out that the method used for plant extraction influences the nature of the components obtained and their relative proportions [29].

In our laboratory, we also studied the essential oil of the fresh leaves of *J. glutinosa*. In this case, extraction was performed by distillation in a modified Clevenger apparatus with a water-cooled oil receiver to reduce artifact formation. The oil was examined by GC–MS. On the other hand, an analysis was conducted by direct thermal desorption (DTD) coupled to GC–MS. Thirty-four components were identified in the essential oil. The main constituents were camphor, borneol, caryophyllene oxide, farnesol and bornyl formate. In addition, cadinol and spatulenol were identified by DTD. Many of these compounds were first described in the essences of this species. After comparing the two methods used, it was observed that the DTD technique had the advantage of requiring less sample and allowing the subsequent identification of a larger quantity of volatile compounds by GC–MS [30]. Sometimes, the sesquiterpene derivatives (alcohols and ketones) described in *J. glutinosa* are erroneously included within the group of sesquiterpene lactones; however, they are not, since they do not have a lactonic ring in their structure. The presence in its composition of secondary metabolites with such varied chemical structures suggests that this species could be used for purposes other than the traditional ones [31].

#### 4.1.3. Biological and Pharmacological Activities

Different studies have been carried out to justify the traditional uses of this species; however, despite the wide range of pharmacological activities attributed to it, there is a notable absence of scientific evidence regarding the effects of the plant on different biological functions and there is little conclusive data relating these actions to a particular active principle or groups of active principles. The biological and pharmacological activities reported for both crude extracts and different isolated compounds are described below.

##### Crude Extracts

Valero et al. evaluated the therapeutic usefulness of the ethanolic extract of *J. glutinosa* in an experimental model of ulcerative colitis induced by 2.5% DSS (sodium sulphate and dextran) in comparison with sulfasalazine as the reference drug and identified the chemical components with anti-inflammatory and/or antioxidant effects. The extract was able to scavenge superoxide radicals, inhibit 5-LOX (5-lipooxygenase) and reduce nitric oxide (NO) and tumor necrosis factor-α (TNF-α) levels. A reduction in myeloperoxidase activity, interleukin-6 (IL-6) and iNOS levels, as well as cyclooxygenase-2 (COX-2) expression were also observed. The results suggest that the phenolic compounds present in the ethanolic extract contribute to the antioxidant and anti-inflammatory effects of *J. glutinosa* [32].

Natural antioxidant compounds play important roles in human health by exerting protective effects against oxidative stress, aging and neurodegenerative diseases. In order to study the phenolic profile and antioxidant activity of aqueous and methanolic extracts of *J. glutinosa*, these extracts were analysed before and after in vitro digestion to simulate human digestive conditions and to evaluate how this process affects the bioavailability of phenolic compounds. The identification and quantification of phenolic compounds present in the extracts was performed by HPLC–MS (HPLC–Mass Spectrometry). The most abundant components were the dicaffeoylquinic acids, representing more than 90% of the total phenol content. Antioxidant activity was determined by spectrophotometric methods. The authors found statistically significant differences in all parameters except for the total phenolic content of both extracts. After in vitro digestion, the number of phenolic compounds, as well as the antioxidant activity, were lower in the case of the aqueous extract. Although digestion reduced the number of phenols and the antioxidant action, the authors suggest that the plant has great potential for its antioxidant properties and due to its content of bioactive phenolic compounds [33].

The antioxidant effect and neuroprotective activity of an ethanolic extract of *J. glutinosa* was studied in vitro, evaluating the ability to inhibit certain enzymes involved in the metabolism and function of acetylcholine, serotonin, dopamine, adrenaline and noradrenaline. These enzymes were monoamine oxidase A, tyrosinase and acetylcholinesterase. On the other hand, neuroprotective activity was evaluated in vivo, in the modified nematode *Caenorhabditis elegans*, as a model of Alzheimer’s disease, in which stress resistance, life expectancy and amyloid toxicity were studied. The extract reduced juglone-induced oxidative stress in nematodes, increased life expectancy and prevented worm paralysis. The paralysis in this nematode model simulates the neurodegeneration and neuronal dysfunction that occurs in the brains of patients with Alzheimer’s disease due to the accumulation of amyloid plaques. Several assays related reactive oxygen species with the development of this disorder, since they are known to foster the production and deposition of amyloid peptides, as well as the hyperphosphorylation of tau protein. In addition, oxidative stress is linked to ageing, so the extracts of *J. glutinosa*, as well as some of its bioactive components, could potentially be useful in the prevention of diseases associated with cellular aging and oxidative stress [34].

In 2022, Mohammed et al. conducted a study on the chemical composition, antioxidant, cytotoxic and antimicrobial activities of the aqueous extract of *J. glutinosa*, harvested in Libya. The total content of phenolic compounds in a hydroalcoholic extract (CH_3_OH/H_2_O 70/30) was determined by LC–MS (Liquid Chromatography coupled with MS). Methoxylated flavonoids and caffeoylquinic acids were identified. Antioxidant activity was evaluated by two complementary methods: the ABTS assay, which uses 2,2′-azino-bis(3-ethylbenzothiazoline-6-sulfonic acid, following the protocol described by Arnao (2001), and the ORAC (Oxygen Radical Absorbance Capacity) assay, based on oxygen radical scavenging capacity and FRAP (Ferric Reducing Antioxidant Power) assay [35,36]. Antiproliferative activity was evaluated in breast adenocarcinoma (MCF-7), hepatocellular carcinoma (HepG2) and pancreatic cancer (PANC-1) cell lines. Antibacterial activity was evaluated using a modified Kirby–Bauer disk diffusion susceptibility test procedure [37]. The authors found differences in phenolic content between the Libyan and Spanish species, which could be attributed to the effects of environmental conditions and soil composition, but also to variations in phenolic compound determination protocols. The extract showed significant antioxidant activity in all three methods used. On the other hand, although the extract showed some level of cell growth inhibition in tumor cell lines, the cytotoxic activity was considered weak compared to doxorubicin and no antimicrobial activity was detected, so the results do not support its use as an antimicrobial agent [38].

The potential of *J. glutinosa* as a dietary additive for aquatic animals has been investigated for its biological and antioxidant activities. These assays were carried out with aqueous and ethanolic extracts and evaluated the effects of diets with different concentrations of this plant on sea bream (*Sparus aurata* L.), observing significant improvements in several immunological and antioxidant parameters, as well as in gene expression related to inflammation and oxidative stress. The authors suggest that the inclusion of this plant in fish feed could enhance health and resistance to stress [39].

Although this species is widely used in some regions of the Iberian Peninsula to treat gastrointestinal issues, scientific evidence regarding its effects on the digestive system is limited. In a study designed to assess the antispasmodic properties of the traditional infusion of *J. glutinosa*, researchers investigated the spontaneous contractions of the rat duodenum in vitro using the plant’s aqueous extract, while also evaluating its in vivo effect on gastrointestinal transit in mice. The results showed that the rock tea extract reduced spontaneous contractions of the duodenal smooth muscle. This inhibitory effect was similar to that observed after the addition of verapamil, an L-type Ca^2+^ channel antagonist. On the other hand, the extract did not modify gastrointestinal transit in healthy mice. However, in a model of colitis induced by dextran sulphate sodium (a substance that causes inflammation of the colon), rock tea extract reversed the increased gastrointestinal transit associated with this condition. The authors conclude that duodenal smooth muscle was relaxed through L-type Ca^2+^ channels by rock tea extract, normalising peristalsis in a colitis sample. These results could support the traditional use of *J. glutinosa* as an antispasmodic in patients with gastrointestinal disorders [40].

In traditional medicine, rock tea is also known for its antihypertensive effect and cardiovascular protective properties; nevertheless, the underlying way it causes a hypotensive action is unknown. Valero et al. studied the vasorelaxant effects of *J. glutinosa* in rat aorta and found that the ethanolic extract produced a relaxation of aortic smooth muscle, which could be explained mechanistically by its inhibitory action on L-type Ca^2+^ channels. The inhibitory effect was similar to that produced by verapamil. Moreover, the extract also blocked the contraction produced by α1-adrenergic receptor activation caused by phenylephrine. The authors note that a limitation of the study is that it is currently unknown whether consumption of rock tea provides effective plasma concentrations of the active constituents to produce an observable antihypertensive effect [41].

Protozoan diseases continue to represent a serious public health problem in emerging countries, and their impact on global health may have been underestimated. Despite being old diseases, they continue to affect a substantial percentage of people in the world, particularly in less developed countries. The potential of natural products from higher plants used in traditional medicine has been widely explored [42]. The *Asteraceae* family, with a diverse range of chemical compounds, has shown remarkable potential for the development of new antiprotozoal therapies [43,44].

Bioactive compounds derived from *Asteraceae* plants, such as flavonoids, terpenes and other secondary metabolites, could be a candidate source for combating this type of disease. In the specific case of *J. glutinosa*, there are few studies in which its antiprotozoal activity has been studied, and all of them are in vitro studies. In a preliminary screening, the acetone extract of the aerial part of this species showed antiprotozoal activity against *Entamoeba histolytica* and *Leishmania donovani.* Leishmanicidal activity was evaluated on *Leishmania donovani* (promastigote forms) in liquid medium (RPMI 1640) and the amoebicidal activity was measured on *Entamoeba histolytica* (Rahman strain) in Jones liquid medium [45].

In a study aimed at evaluating properties of *J. glutinosa* extract on metabolic disorders including enzymatic inhibition bioassays of lipase, α-glucosidase and fatty acid amide hydrolase were performed in cell-free systems. The assay showed the inhibition of lipase, α-glucosidase and fatty acid amide hydrolase [46].

##### Isolated Compounds

Sesquiterpene compounds of the eudesmanolide family are widely distributed metabolites in the plant kingdom and have attracted the attention of numerous researchers for their diverse range of biological activities, including its antifeedant property. This activity is a defensive strategy that many plants and some animals use to protect themselves from predators. They also have cell growth inhibitory activity, as well as plant growth regulatory activity.

Given the antiprotozoal activity shown in previous studies by the acetone extract of *J. glutinosa,* and with the aim of investigating the active compounds of the plant, the effect of the sesquiterpene alcohols 5-epi-kutdtriol and kutdtriol on *Plasmodium falciparum* and *Leismania donovani* was evaluated. (promastigote forms) The results showed that only kutdtriol (**12**) was active against both parasites at a concentration of 250 μg/mL [47].

Among the pharmacological actions described for sesquiterpenes is their anti-inflammatory activity. COX and LOX are key enzymes in inflammation and are therefore pharmacological targets in the search for new therapeutic compounds, since they are responsible for the production of prostaglandins, leukotrienes and thromboxanes. The *J. glutinosa* sesquiterpenes, lucinone (**13**), glutinone (**14**), 5-epi-kutdtriol (**11**) and kutdtriol (**12**), were studied in vitro for their anti-inflammatory properties in cellular systems that generate metabolites from 5-LOX and COX. None of the compounds showed an inhibition on leukotriene C4 (LTC4) production in mouse peritoneal cells stimulated with calcium ionophore (A23187); however, all compounds inhibited prostaglandin E2 (PGE2) release in mouse peritoneal cells, although with lower potency than indomethacin. Glutinone exerted an inhibitory effect on the COX pathway, whereas lucinone, kutdtriol and 5-epi-kutdtriol only showed significant effect on PG-synthase activity. Only glutinone showed a significant reduction in calcium ionophore-induced thromboxane B2 (TXB2) release, but to a lesser extent than a non-steroidal anti-inflammatory drug. The conclusions are that, although the isolated sesquiterpenes possess anti-inflammatory activity, their effectiveness is inferior to that of the reference drugs. The authors identify some structural features as responsible for the inhibitory activity shown on arachidonic acid metabolism. Kutdtriol and 5-epi-kutdtriol differ in terms of stereochemistry (Figure 4). Alteration of the *cis* to *trans* substitution at position 5 is a positive requirement for PG-synthase inhibition. All compounds have a methyl group at position 10, which could be an important functional group for PG synthase activity. On the other hand, the presence of an aliphatic chain at C-10 of glutinone could be a necessary requirement for TX-synthase activity since, when it is blocked, the effect disappears. The results suggest that selective inhibition of the PG-synthase pathway is the main target of action of the sesquiterpenes studied, although glutinone produces moderate inhibition of TX-synthase. However, since TXB2 synthesis from arachidonic acid depends on both COX and TX-synthase activity, the release-dependent suppression of TXB2 release can also be explained by COX inhibition due to glutinone [48].

Las Heras Etayo et al. (2021) conducted a literature review, based on clinical trials, in order to relate the biological activity attributable to the plant with the presence of some chemical compounds previously identified in *C. glutinosus*, namely glutinone, lucinone, borneol, kutdtriol, camphor, kaempferol and quercetin. The authors also conducted an in silico modelling study to anticipate the possible effects of the active compounds of *C. glutinosus.* These experiments enabled us to link certain therapeutic targets reported for this species to the presence of its major components. However, further research is required to validate the findings [31].

### 4.2. Chiliadenus montanus

*C. montanus* (Vahl.) Brullo [*J. montana* (Vahl) Botsch., *Chrysocoma montana* (Vahl.) Symb., Inula conyzoides DC., *Linosyris montana* DC., *Varthemia montana* (Vahl.) Boiss, *Varthemia conyzoides* Boiss] [6], known as Heneida [49], is a medicinal plant native to the Sinai Peninsula [50]. It grows in alkaline soils in the desert east of the Nile, on the Sinai Peninsula in Egypt, the El-Tih desert east of the Suez Canal, in Arabia, on the Mediterranean coast and in Palestine (Figure 1B) [49,50]. The Sinai Peninsula is an epicentre of medicinal plants in the Arabian Desert, with active constituents that serve as a focal point for ecologists, taxonomists and phyto-chemists worldwide. It is home to indigenous medicinal plants used for centuries to treat various diseases, and *C. montanus* is one of the most prominent [50,51]. This plant is considered a vulnerable species due to its restricted habitat, intensive medicinal use and commercial exploitation [52].

A small, yellowish-green shrub with glandular and pubescent features, growing up to 60 cm tall, the stems are highly branched, and the leaves are sessile, oblanceolate or oblong-elliptic, covered with glandular hairs on both surfaces, with wavy edges and an acute or obtuse tip. The plant produces numerous solitary capitula arranged in a loose panicle. The peduncles are bracteate, and the involucre is campanulate, with phyllaries in 3–4 layers that are loosely overlapping, glandular-hairy, and yellow, with the outer ones partially greenish at the rounded apex. The inner phyllaries are scarious and longer. The achenes are densely hairy, and the pappus is reddish–brown (Figure 2B) [50].

#### 4.2.1. Traditional Uses

*C. montanus* is broadly employed in historical Egyptian medicine, highly appreciated among traditional healers and Bedouins. For centuries it has been recognized as an important medicinal plant, whose knowledge and uses have been passed down from generation to generation from the ancient Egyptians to the present day. It is used in infusions for the treatment of various diseases, including “kidney symptoms, chest disorders, diarrhoea and stomach pain” [49,51,53,54].

#### 4.2.2. Phytochemistry

Numerous papers have been published on the existence, in *C. montanus*, of components of different natures, including volatile oils, flavonoids, monoterpenes, diterpenes, triterpenes, sesquiterpenes and sterols. In relation to flavonoids, different aglycones have been identified in the aerial part of *J. montana*, such as quercetin and 6-hydroxykaempferol methyl ethers, as well as quercecetagetin 3,5,6,7,3′-pentamethyl ether (**17**) [55]. In the chloroplast extract of the aerial part, the presence of 3, 6, 7, 3′, 4′-pentamethoxy quercetin (artemetin) (**19**), 3, 6, 7, 3′-tetramethoxy quercetin (chrysosplenetin) (**20**), 3, 6, 3′, 4′-tetramethoxy quercetin (**21**), and 3, 6, 7-trimethoxy quercetin (**22**) has been described, along with 3, 6, 3′-trimethoxy quercetin (jaceidin) (**23**), 3, 6, 4′-trimethoxy quercetin (centaureidin) (**24**), 3, 3′, 4′-trimethoxy quercetin (**25**), 3, 6-dimethoxy quercetin (**26**), 3, 3′-dimethoxy quercetin (**27**), 7, 4′-dimethoxy quercetin (**28**) and quercetin (**18**). The same authors identified three flavonoid glycosides in the ethyl acetate fraction, quercetin-3-O-β-D-galacturonopyranoside (**29**), quercetin-3-O-β-D-glucopyranoside (**3**) and patuletin-7-O-β-D-glucopyranoside (**1**), as well as two phenolic acids in the n-buthanol fraction, 3,5-dicaffeoyl-quinic acid (**31**) and caffeic acid (**32**), and two flavonoid glycosides, quercetin-3-O-α-L-rhamnopyranoside (quercitrin) (**30**) and quercetin-3-O-β-D-glucuronopyranoside (**7**) (Table 3) (Figure 5) [54]. The presence of centaureidin (**24**) in the aerial part of *C. montanus* has been previously described [56].

Other methoxylated flavonoids were isolated and identified from the CH_2_Cl_2_/MeOH (1:1) extract of the aerial part: 5,7-dihidroxi-3,3′,4′-trimetoxiflavone (**25**), 5,4′-dihidroxi-3,6,7,3′-tetrametoxiflavone (crisosplenetine) (**20**), centaureidine (**24**), 5,7-dihidroxi-3,6,3′, 4′-tetrametoxiflavone (bonanzine) (**21**), 5,3′,4′-trihidroxi-3,6,7-trimetoxiflavone (crisosplenol-D) (**22**) y 5-hidroxi-3,6,7,3′,4′-pentametoxiflavone (artemetine) (**19**) (Table 3) (Figure 5) [53,62].

Subsequently, other authors also identified some of these methoxylated flavonoids in a methanolic extract [63].

Within the terpene group, eight metabolites were identified from the CH_2_Cl_2_/MeOH (1:1) extract of the aerial parts, two of which were isolated as free acids. These compounds include 3-oxo-γ-costic acid β-D-glucopyranoside ester (**33**); 3β-methoxy isocostic acid (**34**); 3α-methoxy isocostic acid (**35**); eudesm-11,13-ene-1β,4β,7α-triol (**36**); 3,6,7-trihydroxy-11-methoxy-3,7,11-trimethyldodeca-1,9-diene (chiliadenol A) (**37**); 3-hydroxy-3,7,11-trimethyl-1,6-dodecadien-9-one (chiliadenol B) (**38**); 3-hydroxy-3,11-dimethyl-6β,9α-epidioxy-dodeca-1,7(14),10-triene (chiliadenol C) (**39**); and 3-hydroxy-3,11-dimethyl-6α,9α-epidioxy-dodeca-1,10,7(14)-ene (chiliadenol D, an epimer of chiliadenol C) (**40**). (Table 3) (Figure 6) [57,64].

In a phytochemical analysis of the CH_2_Cl_2_-MeOH (1:1) extract obtained from the leaves, jasomontanone (**41**), an eudesmanolide-type sesquiterpene with a new 5/7-membered bicyclic ring system, was isolated and identified. The compound was identified as (3aR*, 6R*, 8aR)-3a-(hydroxymethyl)-6-(2-hydroxy-propan-2-yl)-8α-methyl-octahydrazulen-4(5H)-one [58]. From the leaves, jasonone (4-hydroxy-5,10-dimethyl-octahydro-azulen-8-one) was also obtained by extraction with CH_2_Cl_2_ (**42**) [54]. Montanone (**43**), an isoiphionane sesquiterpene, was isolated from the aerial part [60].

From the aerial part of *C. montanus*, Elhady et al. also isolated four sesquiterpenes: intermedeol (**44**), 5α-hydroperoxy-β-eudesmol (**45**), eudesm-11,13-ene-1β,4β,7α-triol (**36**) and 1β,4β,7β,11-tetrahydroxyeudesmane (**46**) [61].

Among the diterpenes present in the aerial part, jasonin-a has been isolated (1E)-2-[(2 S)-1,2,5-trimethylbicyclo[3.2.1]octan-8-yl]vinyl bencene-3-carboxylic acid (**47**), jasonin-b ([3-((2 S, 5 S)-1,2, 5-trimethylcycloheptanyl)propyl]bencene-3-carboxylic acid) (**48**) amdasonin-c ([(1 E)-3-((7 R -1,7-dimethyl-4-methylencycloheptanyl)prop-1-enyl]bencene-3-carboxylic acid) (**49**) (Table 3) (Figure 6) [56].

The components of the essential oil were also studied by gas chromatography–mass spectrometry (GC–MS). Fifty-eight components were characterized, of which the main ones were borneol, camphor, chrysanthemol, bornyl acetate, 1,8-cineole, and intermedeol [65].

#### 4.2.3. Biological and Pharmacological Activities

Biological and pharmacological activity studies conducted so far have shown that *C. montanus* has antioxidant, anti-inflammatory, antimicrobial, cytotoxic, antiatherogenic and antidiabetic activity, as well as potential to reduce fat accumulation and body weight, suggesting a beneficial role in the treatment of obesity. In addition, there are studies that attribute anticholinesterase activity, which would suggest its potential use in therapy for Alzheimer disease [66]. The biological and pharmacological activities reported for both crude extracts and different isolated compounds are described below.

##### Crude Extracts

Numerous studies have shown that *C. montanus* extracts possess significant and promising antioxidant activity, mainly attributed to its polyphenols. The ethanolic extract of the aerial part revealed the presence of flavonoids, phenolic compounds that, as is well known, are able to modulate many enzymes and numerous cellular systems. In addition, they have important anti-hepatotoxic, anti-allergic, anti-inflammatory, anti-osteoporotic and even antitumor and antioxidant effects [67].

In 2008, Hussein suggested that some components of this plant scavenged free radicals, producing antioxidant activity, and protected tissues from lipid peroxidation [68]. *C. montanus* contains high levels of polyphenols, mainly flavonoids, which are excellent antioxidants and iron chelating agents. Flavonoids scavenge reactive oxygen species and free radicals by several mechanisms. In addition, the ability shown by the extract to chelate iron contributes to the preventing of oxidative reactions. In 2010, the same author claimed that the ethanolic extract of the aerial part of this species was able to reduce the harmful effects of iron accumulation in an in vivo model of iron overload. The extract produced a statistically significant decrease in liver and kidney iron levels, serum iron, serum ferritin and serum transferrin [69].

On the other hand, the high content of flavonoids and other phenolic compounds in the ethanolic extract seems to be responsible for the anticholestatic activity detected in a rat model of ethinylestradiol-induced biliary cholestasis. In addition, and in agreement with the previous findings, a significant hepatoprotective and antioxidant activity was observed. The authors of this study highlight the therapeutic potential of this plant in drug-induced biliary cholestasis [70]. Likewise, in an obese rat model, the anti-obesity, anti-atherogenic, anti-diabetic and antioxidant activity of the ethanolic extract was studied. This extract helped maintain levels of reduced glutathione (GSH), glutathione peroxidase (GPx), glutathione reductase (GR), superoxide dismutase (SOD) and catalase (CAT), all important antioxidant factors in the liver, which would normally be reduced due to oxidative stress caused by obesity and a high-fat diet. Anti-obesity action is attributed to an increase in liver energy expenditure, a reduction in fatty acid synthesis and an increase in the utilisation of fat as energy [71].

Eissa et al. (2013) investigated the antioxidant and cytoprotective properties of hydroalcoholic extracts of *C. montanus* under H_2_O_2_-induced oxidative stress conditions in a human astrocytoma cell line. Research on the antioxidant capacity was conducted using the ORAC method, comparing with a vitamin E analogue. The results showed the potent free radical scavenging activity of the extract. Such activity could protect against the damaging action of oxidative species and, consequently, against lipid peroxidation in cell membranes, events that are involved in the pathophysiological development of many diseases related to oxidative stress. Phenolic compounds seem to be mainly responsible for this antioxidant activity, which is closely related to the hydroxyl groups attached to the benzene ring in the flavone structure [72]. Other authors attribute the antioxidant effect not only to the flavonoids, but also to the high content of terpenic compounds, mainly sesquiterpenes [73].

The effect of methanolic extract of *C. montanus* leaves against CCl_4_-induced hepatotoxicity in rats was analysed by measuring several hepatic biochemical parameters. The extract reduced hepatotoxicity by scavenging free radicals and significant improvements were observed in all parameters studied, suggesting a hepatoprotective and antioxidant action [74].

In an attempt to explain the biochemical mechanisms responsible for the effect of aqueous and ethanolic extracts of the aerial part of *J. montana* on protein biosynthesis in rat hepatocytes, Hussein et al. (2020) conducted in vivo research on the effects of the administration of these extracts on hepatic and renal tissues. In the liver of treated rats, the extracts promoted an increase in protein, RNA and DNA content. These authors suggest that certain components of the extracts, particularly flavonoids, could promote hepatic synthesis of nucleic acids and proteins by reducing the activity of plasma transaminases and alkaline phosphatase, which could indicate a protective effect on hepatic cells, maintaining the integrity of hepatic and lysosomal cell membranes [75]. As previously mentioned, these extracts have potent antioxidant effects, acting as free radical scavengers and inhibitors of LDL oxidation, and the relationship between this antioxidant action and the hepatoprotective activity shown by flavonoids has been demonstrated. The antidiabetic, hypolipidemic and antioxidant effect of aqueous extract of *J. montana* was studied in rats with alloxan-induced diabetes. Diabetic rats treated with the extract showed a decrease in blood glucose levels, an improvement in lipid profile and a reduction in insulin and total protein levels. On the other hand, treatment with aqueous extract of *J. montana* in diabetic rats was beneficial to the morphological changes produced in liver, kidney and pancreas in diabetic rats, so the authors suggest that this species exerts a powerful protective and tissue regenerative effect, which could be related to its antioxidant activity [76].

Administration of *C. montanus* extract to rats with streptozotocin-induced diabetes showed a reduction in blood glucose levels. The authors attribute this to increased glucose uptake in skeletal muscle and adipocytes, as well as increased insulin secretion in pancreatic cells. Histopathological analysis showed an increase in the number, size and volume of islets of Langerhans, as well as an increased cell density within the islets [77]. These findings are consistent with a previous study that claimed that *C. montanus* had antihyperglycemic properties in alloxan-induced diabetic rats [62]. *C. montanus* extract could exhibit an antidiabetic action through several mechanisms, including inhibition of α-amylase, an enzyme that hydrolyses the α-linkages of starch to produce glucose and maltose [77,78,79].

*J. montana* is also attributed to have anti-inflammatory properties. As early as 2007, it was suggested that sterols and triterpenes were the causative agents of these properties by inhibiting the production of proinflammatory cytokines [59].

In relation to antimicrobial activity, the essential oil showed antibacterial activity against *Bacillus subtilis.* Marked antifungal activity against *Trichophyton mentagrophytes*, *Cryptococcus neoformans* and *Candida albicans* was also found [65]. Al-Howiriny et al. demonstrated that petroleum ether and chloroform extracts of the aerial part of *J. montana* showed activity against *Bacillus subtilis*, *Staphylococcus aureus*, *Mycobacterium smegmatis* and *Candida albicans*, while the chloroform extract showed activity only against *Bacillus subtilis* and *Staphylococcus aureus* [56].

The essential oil of the aerial part of *C. montanus* extracted by hydro-distillation was evaluated for its potential action against H_2_O_2_-induced oxidative stress in human astrocytoma U373-MG cells. The chemical composition of the essential oil was analysed using GC–MS, identifying that the main compounds were monoterpenes and sesquiterpenes, both oxygenated. In vitro evaluation of the antioxidant properties of the essential oil using the ORAC assay revealed a low protective activity against oxidative stress [80].

The hydroalcoholic extract of *C. montanus* showed positive results against poliomyelitis virus (POLIO) with high antiviral activity, confirming the properties attributed in Egyptian folk medicine. The authors relate the antiviral activity to its content in methoxylated flavones, since some 3-methoxyflavones previously isolated, as well as their synthetic derivatives, showed interesting antiviral activity against picornavirus, including POLIO virus [81,82].

Recently, the cytotoxic effects of methanolic and petroleum ether extracts of *J. montana* have been evaluated in colorectal cancer cell lines (HCT-116 and Caco-2), showing a reduction in cell growth rate, as well as a reduction in the expression of some cell differentiation markers in the case of the two extracts, results that can be attributed to both sesquiterpenes and methoxylated flavonoids [83].

Other biological or pharmacological activities have also been described for extracts of *J. montana.* One is the anticholinesterase activity, attributable to the high content of sesquiterpenes and flavonoids in the ethanolic extract. This extract has been shown to have a neurotrophic and anti-amyloidogenic effect, which has led to its possible used in the pharmacological therapy for Alzheimer disease [84].

In addition, antiplatelet aggregation and hypotensive activity has been described for the aqueous extract of *J. montana* [85].

##### Isolated Compounds

Flavonoids are well known for their antioxidant properties. This activity is markedly influenced by the number and position of the hydroxyl groups in the flavone structure, as well as by the degree of conjugation of their rings. The numerous hydroxyl groups in their structure, the presence of methoxyl groups, which increase lipophilicity, and the existence of a double bond and a carbonyl function in the central ring, which increase reactivity by favouring electron conjugation and delocalisation, are essential to their antioxidant action (Figure 5) [53,86]. In the specific case of quercetin (**18**), the most common flavone aglycone in *J. montana*, the main mechanism of its antioxidant action seems to be the donation of hydrogen atoms [87].

Among the multiple pharmacological activities exhibited by flavonoids, their activity against several types of cancer through the modulation of apoptosis stands out [88]. Jaceidin (5,7,4′-trihydroxy-3,6,3′-trimethoxyflavone) (**23**), a methoxylated flavone, showed powerful in vitro cytotoxic activity against liver cancer (HepG2) and breast cancer (MCF-7) cell lines. However, the compound 5,7-dihydroxy-3,3′,4′-trimethoxyflavone (**25**), despite its structural similarity to jaceidin, showed less inhibition of these cell lines, which could be explained by the different configuration and arrangement of the flavonoid B-ring functional groups (Figure 5), since this configuration is decisive for the elimination of free radicals and the inhibition of lipid peroxidation. On the other hand, in order to determine its antitumor effect in vivo, a mouse model of Ehrlich ascites carcinoma (EAC) was used to evaluate the plasma levels of vascular endothelial growth factor (VEGF), as well as several targets of free radicals which could explain its antitumor effect. The results revealed a reduction in tumor weight and an improvement in the histological picture of the tumor, as well as a decrease in oxidative stress. The authors of this work suggest that jaceidin acts as a selective inhibitor of VEGF-mediated angiogenesis. The interaction of jaceidin with target receptors was also studied by molecular docking techniques [61].

The poly-methoxylated and polyhydroxylated flavonoids, 5,7-dihydroxy-3,3′,4′-trimethoxyflavone (**25**) and chrysosplenetin (**20**), present in *C. montanus*, have shown chemoprotective and anticarcinogenic effects, respectively [53,54]. Chrysosplenetin (**20**), centaureidin (**24**) and quercetin-3-O-β-D-glucopyranoside (**3**), isolated from the aqueous extract of the aerial part of *J. montana,* showed a promising cytotoxic effect against the HeLA cervical carcinoma cell line [54].

Moreover, the methoxylated flavone 5,7,4′-trihydroxy-3,3′-dimethoxyflavone (**27**) was shown to negatively regulate LPS-induced inflammatory cytokine expression, including TNFα, IL-1β, nuclear factor kappa B (NF-κB), COX1, COX2 and 5-LOX [63].

The effect of the methoxylated flavones 5,7-dihydroxy-3,3′,4′-trimethoxyflavone (**25**), bonanzin (**21**) and artemetin (**19**), obtained from CH_2_Cl_2_:CH_3_OH (1:1) extract on the induction of the chemo-preventive enzyme NQO1 [(NAD(P)H: Quinone Oxidoreductase 1]. NQO1’s activity is an indicator of the antioxidant capacity of cells, which is measured to understand the toxicity of certain compounds and their potential to induce protective responses in cells. Only the former showed a pronounced dose-dependent induction of NQO1. The other two compounds showed no activity in the concentration range tested, suggesting the importance of substitutions on the A-ring of flavonoids for enzyme-inducing activity (Figure 5) [53].

The other group of compounds of *C. montanus* with interesting biological activities is terpenes. Isolated sesquiterpenes have shown certain cytotoxicity against human colon carcinoma (Caco-2) and human cervical cancer cells [54,64]. The 3-oxo-γ-costic acid, aglycone de 3-oxo-γ-costic acid β-D-glucopyranoside ester (**33**), has antiproliferative activity against human colon and lung cancer cell lines (Caco-1 and A549, respectively). In addition, four sesquiterpenes were obtained from it by fungal transformation using *Athelia rolfsii*, one of which also exhibited antiproliferative activity against the A549 cell line. This is an example of how fungal-mediated chemical transformations can lead to new and potent active agents, in some cases with improved characteristics relative to their precursors [64].

The potential use of *C. montanus* in different industrial areas, as well as in the mitigation of environmental pollution, has been reported. The extensive use of cellulosic and lignocellulosic materials has generated a significant accumulation of waste, contributing to environmental pollution. A Gram-positive endophytic bacterium identified as *Lysinibacillus xylanilyticus* was isolated from *C. montanus*, which is capable of breaking down cellulose [89].

The study of the allelopathic characteristics of *C. montanus* revealed that ethyl acetate and butanol extracts obtained from the aerial part show significant inhibitory activity against several weed species, including *Convolvulus arvensis* and *Calystegia inflata*, as well as other species such as *Portulaca oleracea* and *Arabidopsis thaliana.* This may suggest a potential use as a natural herbicide for weed control. The compounds with phytotoxic activity belong to the group of flavonoids and sesquiterpenes and among the most active were methoxylated flavones. The authors suggest that allelochemicals from *J. montana* could be used both as natural herbicides and as therapeutic agents for diseases caused by microorganisms [90].

### 4.3. Chiliadenus iphionoides

C. iphionoides (Boiss. and Blanche) Brullo [Blanchea iphionioides (Boiss. and C.I. Blanche) Boiss., Iphiona iphionoides (Boiss. and C.I. Blanche) Benth. and Hook.f. ex B.D. Jacks., Jasonia iphionoides (Boiss. and C.I. Blanche) Botsch., Varthemia iphionoides Boiss. and C.I. Blanche] [6,50] is a medicinal species endemic to the eastern Mediterranean [91]. It grows in rocky places and extreme deserts in most regions of Israel, from Mount Hermon to the southern Negev, as well as in Syria, Lebanon, Jordan and the Sinai Peninsula (Figure 1C) [6].

It is a small perennial shrub, distinguished by its aroma, 20–50 cm high, with a woody base, hairy and sticky aromatic stems, small leaves covered with hairs and glands containing essential oil, which gives the plant its unique aroma. Its flowers are tubular and yellow (Figure 2C). It flowers from September to December [92].

As discussed in Section 2 of this article, *C. iphionoides* was previously classified under the name *V. iphionoides.* Since many relevant studies were published using this earlier nomenclature, it has been decided to incorporate these papers to provide a complete and accurate overview of the research carried out on this species. In these cases, we will refer to it as *V. iphionoides.*

#### 4.3.1. Traditional Uses

*C. iphionoides* has been commonly employed as herbal medicine in pain mitigation, wounds, eye infections, and in cases of urinary retention [92]. It has also been used for gastrointestinal disorders, diabetes, male and female fertility disorders, kidney stones, fever, influenza, depression, stomach pain, nervousness and as an anti-inflammatory [93]. Fresh leaves are used externally, or fresh or dried leaves are administered orally as an infusion, based on the indication.

It is also attributed to have anticancer, antidiabetic, antimicrobial, antioxidant, antispasmodic and antiplatelet action, among others. In addition to its medicinal uses, *C. iphionoides* has been used as a deodorant, cooking spice, condiment and as a sugar-sweetened herbal tea [94].

#### 4.3.2. Phytochemistry

Different flavonoids were separated and identified from the aerial part of *V. iphionoides*, including xanthomicrol (**50**), kumatakenin (**51**), jaceidin (**23**) and 3,3′-dimethoxy quercetin (**27**) (Table 4) (Figure 7) [95].

Subsequently, also from the aerial part, seven other methoxy-flavones were isolated and identified: 5,7,4′-trihydroxy-3,6-dimethoxyflavone (**52**), 5,7,4′-trihydroxy-3,3′-dimethoxyflavone (**27**), 3,5,6,7,8,4′-hexamethoxytflavone (**53**), 5,4′-dihydroxy-3,6,7-trimethoxyflavone (**54**), 5,7,4′-trihydroxy-3-methoxyflavone (**55**), 5,4′-dihydroxy-3,7-dimethoxyflavone (**56**) and 5,4′-dihydroxy-3,7,3′-trimethoxyflavone (**57**) (Table 4) (Figure 7) [96].

**Table 4 plants-14-00205-t004:** Main components isolated in *V. iphionoides*.

Compound	Reference
Flavonoids	
Xanthomicrol (4′,5-dihydroxy-6,7,8-trimethoxyflavone) (**50**)	[95]
Kumatakenin (4′,5-dihydroxy-3,7-dimethoxyflavone) (**51**)	[95]
Jaceidin (**23**)	[95]
3,3′-dimethoxy quercetin (**27**)	[95]
5,7,4′-trihydroxy-3,6-dimethoxyflavone (**52**)	[96]
3,5,6,7,8,4′-hexamethoxytflavone (auranetin-5-methylether) (**53**)	[96]
5,4′-dihydroxy-3,6,7-trimethoxyflavone (**54**)	[96]
5,7,4′-trihydroxy-3-methoxyflavone (**55**)	[96]
5,4′-dihydroxy-3,7-dimethoxyflavone (**56**)	[96]
5,4′-dihydroxy-3,7,3′-trimethoxyflavone (**57**)	[96]
Sesquiterpenes	
3-oxocostusic acid [selina-4,11(13)-dien-3-on-12-oic acid] (**58**)	[97]

A sesquiterpene was isolated and identified from the ethyl acetate extract of the aerial part of *V. iphionoides*, 3-oxocostusic acid [selina-4,11(13)-dien-3-on-12-oic acid] (**58**) [97]. This compound had been previously identified in *Artemisia altaiensis* [98].

The study of the chemical composition of the essential oil of *V. iphionoides* by gas chromatography–mass spectrometry (GC–MS) resulted in the identification of caryophyllene, eudesmol and spatulenol as main components, known for their aromatic and therapeutic properties, among which are anti-inflammatory and central nervous system relaxing effects [99].

The results of a study describing the content and composition of the essential oil of *C. iphionoides* in plants from the Jordan Valley (Jordan) showed the presence of borneol, as the main constituent, as well as 1,8-cineole, α-terpineol, camphor, bornyl formate, terpinen-4-ol and bornyl acetate. Phytochemical analysis of *C. iphionoides* collected in Israel showed a different composition of the essential oil, as these populations were found to be rich in intermedeol, a product not found in Jordanian plants. Chrysanthemic acid and τ-cadinol were also found. In addition, a large variation in essential oil composition was found among the different Israeli populations [100]. Subsequently, in order to evaluate the differences in essential oil composition among the different wild populations of *C. iphionoides*, a study of the endemic species from all over Israel was conducted. The essential oil was obtained by hydro-distillation of the aerial parts of the plants and analysed by GC–MS. Considerable variation was observed in the composition of the essential oils of the different populations, with a significant correlation between chemical composition and geographic location. Three main chemotypes were identified: a camphor/α-pinene/phochienol chemotype, a τ-cadinol/1,8-cineole/trans-chrysanthemol chemotype, and an intermedeol chemotype [101].

#### 4.3.3. Biological and Pharmacological Activities

##### Crude Extracts

There are numerous studies describing the biological activity of *C. iphionoides* extracts. In one of the first published works on this species, the aqueous extract showed limited antibacterial activity against *Escherichia coli*, *Staphylococcus aureus*, *Salmonella typhimurium* and *Bacillus cereus* [95]. In a later publication, the ethyl acetate extract of the aerial part of *V. iphionoides* showed in vitro antibacterial activity against *Staphylococcus aureus*, *Bacillus subtilis*, *Micrococcus luteus*, *Escherichia coli*, *Bacillus cereus* and *Salmonella enteritides.* In contrast, hexane and ethanol extracts were not active.

Masadeh et al. (2013) evaluated the antimicrobial activity of the ethanolic extract of *V. iphionoides*, among other Jordanian medicinal plants, against four types of bacteria: *Enterobacter faecalis*, *Salmonella typhi*, *Staphylococcus aureus*, and *Escherichia coli.* Antimicrobial susceptibility tests were carried out and the minimum inhibitory concentration (MIC) was determined. The ethanolic extract of *V. iphionoides* was the most active against *E. faecalis* [102].

Al-Dabbas et al. conducted research on the antioxidant and α-amylase inhibitory compounds present in the aerial parts of *V. iphionoides.* They isolated several bioactive compounds and evaluated their antioxidant capacity by free radical scavenging assays, such as DPPH (2,2-diphenyl-1-picrylhydrazyl), as well as their ability to inhibit the enzyme α-amylase, which is relevant to the control of blood glucose levels. The results showed the presence of flavonoids and other phenolic compounds in the extracts of *V. iphionoides*, with significant antioxidant activity and remarkable ability to inhibit α-amylase. These findings suggest that *V. iphionoides* may be a valuable source of natural compounds useful in the prevention of oxidative stress and in the management of diabetes, highlighting its therapeutic potential and its relevance in the development of plant-based treatments [96].

Other authors also investigated the antioxidant, antimicrobial and cytotoxic properties of *C. iphinoides* in order to support its traditional use. Extracts of different polarity (methanol, acetone, hexane and water) were analysed. Antioxidant activity was determined by the DPPH method. Antibacterial activity was evaluated against six strains: *Pseudomonas aeruginosa*, *Escherichia coli*, *Shegilla sonnie*, *Enterococcus faecium*, *Staphylococcus aureus* and methicillin-resistant *Staphylococcus aureus.* The antifungal activity against *Candida albicans* and *Epidermatophyton floccosum* was also determined. Antiproliferative and apoptotic activity was assessed by colorimetric methods. All four extracts, aqueous, methanolic, acetonic and hexane, showed antioxidant activity. The acetonic extract showed the highest antimicrobial potency; next was the hexane and methanolic extracts, whereas the aqueous extract only showed antimicrobial activity against *Shegilla sonnie.* The acetone and hexane extracts showed potent antiproliferative properties, as did the methanolic extract [103].

The first study demonstrating the hypoglycemic effect of *C. iphionoides* in in vitro models of diabetes and in rats with streptozotocin-induced type 2 diabetes was conducted in 2011. In diabetic animals, the aqueous extract of the aerial part reduced blood glucose levels by 70% after one hour. In healthy rats, the extract also reduced blood glucose four hours after treatment. Factors that modulate blood glucose levels, such as insulin production and glucose utilisation by muscle and adipose tissue, were investigated. In cellular models, *C. iphionoides* extract increased insulin secretion in pancreatic cells and glucose uptake in skeletal muscle and adipocytes. In rats, the extract reduced blood glucose levels in both oral tolerance tests and long-term feeding studies. Treatment with the extract also significantly decreased total blood lipid levels and hepatic cholesterol. The data obtained showed that the antidiabetic effects of the plant could be linked not only to amylase inhibition, but also to an increase in insulin secretion and glucose utilisation [104,105].

Kasabri et al. (2011) provided new data on the possible mechanisms involved in the antidiabetic effect shown by the aqueous extract of *V. iphionoides.* They evaluated in vivo and in vitro the plant’s postprandial antihyperglycemic activity on the enzymatic digestion of starch (using α-amylase and α-amyloglucosidase) as a possible mode of antidiabetic action. In vitro, aqueous extract of *V. iphionoides* exerted significant dual dose-dependent inhibition of α-amylase and α-glucosidase, thus the authors suggest that it could improve glucose homeostasis by significantly delaying carbohydrate absorption [106].

A study was conducted to investigate the metabolic effects of *C. iphionoides* consumption in an animal model of obesity and non-alcoholic osteopathy induced by a high-fat diet. Dietary supplementation with a hydroalcoholic extract significantly reduced the weight gain induced by the obesogenic diet. Likewise, fat accumulated in the liver and adipose tissue were reduced for the animals which consumed *C. iphionoides.* Reduction of hepatic steatosis was demonstrated histologically, and by plasma levels of liver enzymes and lipid profile, which indicated a favourable evolution of liver function and lipid metabolism, as well as increased insulin sensitivity [107].

Numerous natural products obtained from plants are currently used as cytotoxic agents (colchicine, vinca alkaloids, paclitaxel, among others), so this field has always been a source of research. The first references to the cytotoxic activity of *C. iphionoides* date back to 1999 [98]. Subsequently, Al-Dabbas et al. studied the in vitro cytotoxic activity of extracts of different polarity of *V. iphionoides* on human leukaemia cells (HL-60) and suggested its anticancer potential [96].

In 2013, Elbadry et al. investigated the in vitro cytotoxic effect of acetone, ethanol, ethyl acetate, petroleum ether, methanol and aqueous extracts of the aerial part of *C. iphionoides* on HeLa cervical carcinoma and MCF-7 human breast adenocarcinoma cell lines. Every extract studied exhibited cytotoxic effects against the HeLa cell line, with the aqueous extract showing the least activity. On the other hand, acetone and methanol extracts showed a very strong cytotoxic activity against the MCF7 cell line, while ethyl acetate and petroleum ether extracts showed a strong cytotoxic activity, but with a higher percentage of cell viability, compared to the two previous examples [108].

In the same year, Thoppil et al. evaluated the antitumor activity of extracts of *V. iphionoides*, among other species, against human hepatocellular carcinoma HepG2 cells. They investigated its effect on cell viability by MTT [3-(4,5-dimethylthiazol-2-yl)-2,5-diphenyltetrazolium bromide] assay, observing induction of apoptosis and morphological changes in neoplastic cells. *V. iphionoides* extract showed significant cytotoxic activity, reducing cell viability in a dose-dependent manner. Induction of apoptosis was also found in cells treated with the extract, as evidenced by DNA fragmentation and the formation of apoptotic bodies [109].

##### Isolated Compounds

Flavonoids **50**, **51**, **23** and **27**, obtained from the aerial part of *V. iphionoides* [95], were studied from the point of view of their antispasmodic activity on rabbit intestinal smooth muscle. Compounds **50** and **23** showed dose-dependent activity [110]. In another study, the antimicrobial activity of flavonoids present in the aqueous extract was evaluated against *Escherichia coli*, *Staphylococcus aureus*, *Salmonella typhimurium* and *Bacillus cereus.* Compound **51** was active against the fungi *Fusarium solani* and *Candida tropicalis*, while compounds **27** and **50** showed higher activity against *Aspergillus parasiticus*, *Candida tropicalis and Fusarium solani* [95].

In subsequent research, the antiplatelet activity of aqueous and ethanolic extracts, essential oil and pure flavonoids (**50**, **51**, **23** and **27**) isolated from the aerial part of *V. iphionoides* was evaluated. The activity was determined by optical aggregometry using collagen and ADP (Adenosine Diphosphate) as aggregation inducers. Compound **50** was the more potent in both cases, while compound **23** did not show significant antiaggregant activity. Neither the alcoholic extract nor the essential oil showed anti-aggregation activity with either of the two aggregation inducers. The aqueous extract showed a dose-dependent anti-aggregation activity against aggregation induced by both collagen and ADP. It is concluded that the aqueous extract of *V. iphionoides* thus contains compounds with significant antiplatelet aggregation activity [111].

In 2006, the antioxidant activity of certain extracts of *V. iphionoides* was evaluated. Since, as previously mentioned, the antioxidative capacity of many plants is linked to the phenolic compound content, the relationship between this activity and the phenolic compound content of various extracts of this species was evaluated, and it was observed that the content was highest in the ethanolic extract, followed by the aqueous extract, ethyl acetate, chloroform and hexane (in decreasing order). The activity of the isolated products was also studied. Flavonoids **52**, **27** and **57** showed a pronounced free radical scavenging capacity [96].

In 2005, these same authors reported the potent antimicrobial activity of sesquiterpene 58 against six different types of bacteria (*Bacillus subtilis*, *Staphylococcus aureus Escherichia coli*, *Micrococcus luteus*, *Salmonella enteritides* and *Bacillus cereus*) [97].

### 4.4. Chiliadenus lopadusanus

*C. lopadusanus* Brullo [*Jasonia lopadusana* (Brullo) M. Pardo and R. Morales] is an herbaceous, perennial plant endemic to the island of Lampedusa, the hugest island of the Pelagian archipelago in Italy, where it grows spontaneously in rocky and sunny places (Figure 1D) [6,112].

It can reach a variable height depending on environmental conditions. Its leaves are alternate, linear or lanceolate, with glandular trichomes of different types on both epidermises. Inflorescences are yellow flower heads typical of the Asteraceae family, which are grouped in clusters. Each capitulum has ligulate flowers at the edge and tubular flowers in the middle (Figure 2D). It has a typical camphoraceous odour [113].

#### 4.4.1. Traditional Uses

There are not many published items of data on the traditional uses of *C. lopadusanus*, but based on its bioactive components, such as farnesane-type sesquiterpenes, its medicinal potential in the treatment of bacterial skin infections and wounds can be inferred [112]. In addition, although not extensively documented, it is likely that, like other plants in the *Asteraceae* family, *C. lopadusanus* has traditionally been used for its anti-inflammatory properties to relieve muscle and joint pain, as well as to treat digestive problems such as dyspepsia and upset stomach. An exhaustive review of the ethnobotanical and pharmacological literature would be necessary to find direct references to these properties.

#### 4.4.2. Phytochemistry

The first published study on the chemical composition of the essential oil of the aerial part of *C. lopadusanus* was that of Sacco and Maffei, which provided a preliminary identification of its constituents [113]. In 2013, the components isolate in the essential oil extracted by hydro-distillation of the aerial part (leaves and flowers) of this species were analysed by GC–MS and compared with the results obtained previously. A total of 98 components were identified in the essential oil from the leaves and 83 in the oil from the flowers. The most common compounds in both oils were monoterpenes and sesquiterpenes, both oxygenated. The principal constituent was camphor (in leaves and flowers), and the next in leaves was torreyol, τ-cadinol and 1,8-cineole, but in flowers it was the sesquiterpenes τ-muurolol, torreyol and τ-cadinol [114]. Several of the identified compounds could play an important role in the antibacterial, antifungal, allelopathic and spasmolytic activity described for this species [115].

Masi et al. (2021) conducted the purification and chemical characterisation of the principal metabolites present in the n-hexane extract of the aerial part of *C. lopadusanus.* The isolated compounds were identified using spectroscopic techniques and compared with data reported in the literature. These compounds were 9-hydroxynerolidol (**59**), 9-oxonerolidol (**60**) (Figure 8) and chiliadenol B (**38**) (Figure 6).

Masi’s work addressed the determination of the absolute configuration of these non-cyclic sesquiterpenes by vibrational circular dichroism and electronic circular dichroism These compounds present analytical challenges due to their flexible structure, which allows the molecules to adopt multiple conformations in solution, complicating their structural analysis, and also due to the absence of rings [112]. The combination of the two circular dichroism techniques allowed the determination of the absolute configuration of non-cyclic sesquiterpenes [116].

#### 4.4.3. Biological and Pharmacological Activities

Masi et al. (2021) conducted research on the antimicrobial effect of *C. lopadusanus* extract against the pathogens, *Staphylococcus aureus* and *Acinetobacter baumannii.* The isolated compounds, particularly the farnesane-type sesquiterpenes, were identified as responsible for this activity. The antimicrobial activity was evaluated by determining the MIC. The crude extracts were able to inhibit biofilm formation and eradicate already preformed biofilms of methicillin-resistant *Staphylococcus aureus* by crystal violet assays. On the other hand, the three isolated farnesane-type sesquiterpenes also showed significant antibacterial activity against *Staphylococcus aureus* and greater effectiveness in inhibiting and eradicating *Staphylococcus aureus* biofilms compared crude extracts. However, neither the crude extracts nor the isolated compounds showed antibacterial activity against *Acinetobacter baumannii* [112].

### 4.5. Chiliadenus bocconei

*C. bocconei* Brullo (*Jasonia bocconei* (Brullo) M. Pardo and R. Morales) [6] (The Global Compositae Checklist), commonly known as *Maltese fleabane* or *tulliera salvagga*, is a plant native to Malta (Figure 1E) and typically thrives in coastal garrigue environments, favouring dry, rocky areas (Figure 2E). Well-suited to arid conditions, it grows in narrow crevices on barren, rocky slopes of cliffs and valleys near the coast, where small amounts of calcareous soil can be found [117].

#### 4.5.1. Traditional Uses

After reviewing the literature, no data on its traditional use have been found, although Buhagiar et al. (2015) refer to some data of this type in a publication by Lanfranco in 1992. *C. bocconei* was employed in Maltese traditional medicine for respiratory problems and inflammatory processes [118,119], although specific documentation is sparse and requires further review to confirm this data.

#### 4.5.2. Phytochemistry

The only published study on the chemical composition of *C. bocconei* refers to the essential oil obtained from the leaves (collected in winter and summer). Buhagiar et al. identified camphor, borneol and τ-cadinol in summer-collected plant and τ-cadinol, camphor, oplopenone, aloaromadendrene and cis-γ-cadinene, in winter-collected leaves. The presence of camphor was lower in winter foliage, which may explain the near absence of odour in this period [119].

There are no studies in the literature on the non-volatile components of this species.

#### 4.5.3. Biological and Pharmacological Activities

There are no published articles on the biological and pharmacological activity of this species, neither of the crude extracts nor of the isolated compounds. Only one work appears in the literature as part of a doctoral thesis, whose general objectives were to study the phytochemical profile and antioxidant effects of four medicinal plants selected from Malta on the basis of ethnopharmacological records. One of these species was *C. bocconei.* These discoveries served as the foundation for bioassays aimed at uncovering the molecular mechanisms responsible for the extracts’ anti-inflammatory effects in human cell models of inflammation. The aerial part of *C. bocconei* was extracted with five different solvents and in each extract the total content of alkaloids, triterpenes and phenols was determined by spectrophotometric assays. Antioxidant activity was studied using the free radical scavenging method. The effect of the extracts on the release of proinflammatory cytokines, TNF-α and IL-1β, was also investigated. Furthermore, studies were conducted to evaluate the impact of the extracts on NF-κB transcriptional activity within human cell models of inflammation. Ethanolic, aqueous, and acetonic extracts of *C. bocconei* were found to significantly inhibit the release of TNF-α and IL-1β induced by lipopolysaccharide (LPS) in phorbol ester-differentiated U937 cells. The acetone extract of *C. bocconei* showed an excellent profile in terms of power and efficacy in inhibiting TNFα release. This represents the first scientific investigation documenting the properties of *C. bocconei* extracts on the inhibition of the release of the IL-lβ and TNFα from human cell lines. Determination of the phytochemical, antioxidant and anti-inflammatory properties of selected Maltese medicinal plants are from the doctoral dissertation, https://www.um.edu.mt/library/oar/handle/123456789/31934, accessed on 10 September 2024).

### 4.6. Chiliadenus antiatlanticus

*C. antiatlanticus* (Emb and Maire) Gómiz [*Jasonia antiatlántica* (Emb. and Maire) Gómiz, *Jasonia antiatlantica* (Emb. and Maire) Gómiz, *J. glutinosa* var. *antiatlantica* Emb. and Maire] is an Asteraceae endemic to southwestern Morocco (Figure 1F). It is characterized by its thickened roots, linear-lanceolate leaves, entire and alternate, up to 45 × 5 mm, subspeciolate, with silky hairs and heterogamous capitula, with peripheral ligulate female flowers (Figure 2F). The plant is weakly hairy and grows on siliceous rocks [120].

Finding specific and detailed information on *C. antiatlanticus* is a challenge, as it is not a widely documented plant in accessible sources. The only data concerning the chemical composition and biological activities of this species were published in 2021. There is no record of data prior to that date. Its chemical composition has been investigated, and the in vitro antibacterial and cytotoxic activity of its essential oil. 27 metabolites were identified in the essential oil, including camphor and derivatives, borneol and camphene, as well as intermedeol, α-pinene and (E)-pinocarveol [121]. Camphor and related metabolites have already been identified as major components of the essential oils of species of the same genus, including *C. montanus*, collected in Egypt [65] and *C. glutinosus*, harvested in Spain [29,30]. They have also been identified in the essential oil of *C. lopadusanus*, endemic to the island of Lampedusa (Italy) [113].

The study of the antimicrobial effect of the essences was positive against 24 Gram-positive strains, obtaining promising results against *Staphylococcus* and *Enterococcus* species. In addition, the essential oil also showed significant toxicity against different tumor cell lines (liver HepG2 and melanoma B16 4A5) [121].

There are no results published on the biological activity of the constituents isolated from this species.

### 4.7. Chiliadenus candicans

*C. candicans* (Delile) Brullo [*Chrysocoma candicans* Delile, *Jasonia candicans* (Delile) Botsch., *Linosyris candicans* (Delile) DC., *Varthemia candicans* (Delile) Boiss.] is a perennial plant that has been widely recognized for its distinctive characteristics and its distribution in Mediterranean regions. After the reclassification of this species, based on taxonomic studies and morphological and genetic analysis, it was placed in the genus *Chiliadenus*, so that the currently accepted name for this plant is *C. candicans* (Cass.). Brullo. The native range of this species is Egypt, the Gulf countries, Libya, Palestine, Sinai, and up to the Arabian Peninsula (Figure 1G) [6].

There are no published data on its traditional medicinal use.

#### 4.7.1. Phytochemistry

Methoxylated flavonoids were isolated from the leaves of this species 6,4′-dihydroxy-3,5,7-trimethoxyflavone (6-hydroxykaempferol 3,5,7-trimethyl ether) (**61**) and 3′-hydroxy-3,5,7,4′-tetramethoxyflavone (quercetagetin 3,5,7,4′-tetramethyl ether) (**62**), besides the already known, luteolin (**63**), isokaempferide (**64**), penduletin (**65**), jaceidin (**23**), and methyl ethers of 6-hydroxykaempferol and quercetagetin [122].

Continuing the phytochemical study and the search for bioactive compounds from this species, these same authors reported the isolation and structural elucidation of seven sesquiterpenes, jasonol (**66**), a tricyclic eudesmane, two 7-epi-eudesmanes, 12-hydroxyisointermedeol (**67**) and 7-epi-ilicic acid (**68**), two guaianolides, lβ-hydroxy-8-epi-inuviscolide (**69**) and 5α-hydroxy-10,14-dihydroinuviscolide (**70**), the iphionane derivative and the cadinane-triol (**71**). They also isolated other compounds already known, such as iso-intermedeol (**72**), pseudo-ivalin (**73**), 8-epi-inuviscolide (**74**) and 5β-hydroxy-4-oxo-ll(13)-dehydroiphionan-12-oic acid (**75**) [123].

Moreover, other sesquiterpenes, as well as the already known constituents confertin (**76**), 4,11(13)-eudesmadien-12-oic acid (**77**), and 11-eudesmen-4-ol (**78**), were isolated and identified in the Et_2_O/CH_3_OH extracts. In addition, two new diol α-methylene lactone antimicrobial agents were identified from NMR, MS and X-ray crystallography data as (4α,5α,8β,10β)-4,10-dihydroxy-1, 11(13)-guaidien-12,8-olide (**79**) and (4α, 5α, 8β,10α)-4,10-dihydroxy-1,11 (13)-guaidien-12,8-olide (**80**), which differ in stereochemistry at the C-10 tertiary alcohol centre (Table 5) (Figure 9) [124].

Hammerschmidt and coworkers (1993) analysed the essential oil from the aerial part of *J. candicans* by GC–MS and identified twenty-one components, of which the main constituent was intermedeol [65].

#### 4.7.2. Biological and Pharmacological Activities

Hammerschmidt et al. investigated the antibacterial and antifungal activity of the essential oil of the aerial part of *J. candicans* and observed a marked activity against *Bacillus subtilis* and an interesting antifungal activity against *Trichophyton mentagrophytes*, *Cryptococcus neoformans* and *Candida albicans* [65].

Ahmed et al. conducted research on the possible therapeutic role of the ethanolic extract of the aerial part of *J. candicans* in the regression of Alzheimer’s disease in experimental rat models. They evaluated the effect on several biochemical and physiological parameters and found anticholinesterase activity and reduction of TNF-α levels, as well as antioxidant and anti-inflammatory properties. The extract also exhibited anti-amyloidogenic potential and neurotrophic effects, significantly reducing amyloid plaque formation in the brain and improving brain insulin-like growth factor-1 (IGF-1) and NF-κB levels [84].

In another study, CH_2_Cl_2_ extract of the aerial part of *J. candicans* significantly reduced cell viability in the drug-sensitive CCRF-CEM leukaemia cell line. The author suggests that this is a promising cytotoxic extract that could be used to combat cancer cells with a multi-drug resistance phenotype through different cell death pathways [125].

There are no results published on the biological activity of the pure constituents isolated from this species.

### 4.8. Chiliadenus hesperius

*C. hesperius* (Maire and Wilczek) Brullo (*Jasonia hesperia* Maire and Wilczek) [6] is endemic to Morocco (Figure 1H). It is a perennial plant, somewhat woody at the base, which grows in rock crevices. It has elliptic-oblong or lanceolate leaves, covered with dense silvery-sericeous indumentum. It has capitula with a yellow ligule (Figure 2G).

Maire and Wilczek [126] described *J. hesperia*, a plant collected in the Tachilla *jbel*, as a new species. Brullo (1979) gives the species as good, although transferring it to the genus *Chiliadenus.* Pardo de Santayana and Morales (2004) provisionally include the species in question as a synonym of *J. rupestris* Pomel, based on the affinity that, according to the authors, both plants present. Subsequently, Gómiz and Morales (2006) confirmed the specific independence of *J. hesperia* from *J. rupestris* [120].

There are no published data in the literature on traditional uses of this species. Neither has it been investigated from a phytochemical point of view, nor from the point of view of its biological activities.

### 4.9. Chiliadenus rupestris

*C. rupestris* (Pomel) Brullo (*J. rupestris* Pomel) [6] is a woody and sticky shrub, which grows in the fissures of limestone rocks. The native range of this species is North Africa, Algeria, Libya and Morocco (Figure 1I). It has obovadolanceolate leaves, sometimes very small, smaller than 1 cm, with involucral bracts of heterogeneous size. Its capitula have reddish ligule, in open inflorescences (Figure 2H) [120].

There are no published data in the literature on traditional uses of this species. Neither has it been investigated from a phytochemical point of view, nor from the point of view of its biological activities.

### 4.10. Chiliadenus sericeus

*C. sericeus* (Batt. and Trab.) Brullo [6] [*J. sericea* Batt. and Trab.= *V. sericea* (Batt. and Trab.) Diels] is a species with a native range in the central Sahara and northwestern Egypt, Algeria, Chad and Libya (Figure 1J) [9].

There are no published data in the literature on traditional uses of this species. Neither has it been investigated from a phytochemical point of view, nor from the point of view of its biological activities.

In Table 6, the main pharmacological activities described for the different species of the *Chiliadenus* genus are shown.

## 5. Discussion

The comprehensive analysis conducted in this review highlights the importance of the genus *Chiliadenus* in ethnopharmacology, underlining its role as a rich and diverse source of bioactive compounds with significant therapeutic potential. Comparative study of the chemical composition and pharmacological activities of the component species reveals significant differences between them, reflecting their diverse therapeutic potential. For example, *C. montanus* stands out for its high concentration of flavonoids and phenolic compounds, responsible for its remarkable antioxidant, anti-inflammatory and anti-diabetic activities. These properties coincide with its traditional use in Egyptian medicine to treat digestive and kidney disorders. The link between its bioactive compounds and the observed therapeutic actions is clear, which allows validation of its ethnopharmacological use in the treatment of various diseases.

In the case of *C. glutinosus*, there is a clear correlation between its traditional ethnopharmacological uses and the biological activities reported in the literature. This plant has been traditionally used to treat a wide range of conditions, including digestive, respiratory and circulatory problems, and for the relief of joint and rheumatic pain. Scientific studies have confirmed several of these applications, highlighting its anti-inflammatory, antioxidant and antispasmodic properties. The anti-inflammatory activity found for *C. glutinosus* extracts could justify their use in the treatment of rheumatic conditions and digestive disorders. In addition, its antioxidant action supports its traditional use in the relief of respiratory and circulatory problems, as these conditions are often associated with oxidative stress.

The same applies to *C. iphionoides.* This species was traditionally employed in Middle Eastern traditional medicine for digestive, inflammatory and respiratory disorders. Scientific studies have supported these uses, demonstrating that it possesses anti-inflammatory, antioxidant, and antimicrobial properties that coincide with its ethnopharmacological applications. Likewise, studies conducted with *C. bocconei* have shown that it possesses antimicrobial and antioxidant properties, which supports its traditional use in the treatment of infections and digestive problems.

In contrast, species such as *C. candicans*, *C. antiatlanticus*, *C. rupestris*, *C. hesperius* and *C. sericeus* show a notable absence of ethnopharmacological data, which could be attributed to limited historical documentation or to the limited use of these plants in traditional medicine. In this regard, *C. candicans* is an interesting exception, since, despite having no documented history of traditional uses, it has been the subject of extensive phytochemical studies. The foregoing suggests that scientific research on this species has been driven more by its geographic distribution or unique morphological characteristics than by prior popular knowledge, highlighting the need to further explore its therapeutic potential.

The correlation observed between traditional uses and biological activities reported for some species underlines the validity of ethnopharmacological knowledge. However, for those species without traditional use data, more detailed ethnographic and pharmacological studies are crucial.

On the other hand, this review also highlights certain gaps in the current research. Despite significant advances in the identification of bioactive compounds, most studies have focused on in vitro assays, leaving a considerable gap in in vivo evaluation and clinical studies. This limitation is critical, as the activity observed in in vitro studies does not always translate into clinical efficacy, underscoring the need for further research in animal models and human trials to confirm *Chiliadenus*’ therapeutic potential.

Finally, the review highlights a persistent problem in the taxonomy of the genus *Chiliadenus*, where confusions and synonymies between species have hindered an accurate characterisation of their distribution and properties. This situation not only complicates the correct identification of species in pharmacological studies, but may also have led to the underestimation of the potential of certain species within the genus.

## 6. Conclusions and Perspectives

This review underscores the promising therapeutic potential of the genus *Chiliadenus*, particularly in relation to its rich array of bioactive compounds such as flavonoids, terpenes, and essential oils. The findings support the traditional uses of these species in folk medicine, especially for their anti-inflammatory, antioxidant, antimicrobial, and antitumor properties, which have been confirmed through in vitro studies.

Despite these encouraging results, several important areas require further investigation to fully realize the therapeutic potential of *Chiliadenus* species. First, in vivo studies and clinical trials are critical to validate the efficacy and safety of *Chiliadenus* compounds. While the in vitro evidence is promising, clinical research is necessary to assess their real-world applications and determine the appropriate dosages and safety profiles for therapeutic use. This is an essential step towards the incorporation of these compounds into modern medical treatments.

Second, the taxonomic classification of *Chiliadenus* must be clarified to avoid any confusion regarding species identification. Correctly identifying the species is crucial for accurately assessing their pharmacological potential and ensuring that future research is based on reliable data. The current ambiguities in species classification could lead to inconsistencies in study results, hindering progress in this field.

Finally, to maximize the therapeutic use of *Chiliadenus*, a more integrative approach is necessary. Future research should combine efforts from multiple disciplines, including chemistry, pharmacology, toxicology, molecular biology, and ecology. Moreover, collaborative studies involving researchers from different regions where *Chiliadenus* species grow naturally would provide valuable insights into their ecological and cultural significance. This interdisciplinary approach will not only strengthen the scientific evidence behind the traditional uses of these plants but also uncover new bioactive compounds with potential clinical applications.

In conclusion, *Chiliadenus* represents an underexplored yet highly promising genus for the development of novel therapeutic agents. With comprehensive, multidisciplinary research, these species could make significant contributions to modern medicine while preserving their cultural and ecological value.

## Figures and Tables

**Figure 1 plants-14-00205-f001:**
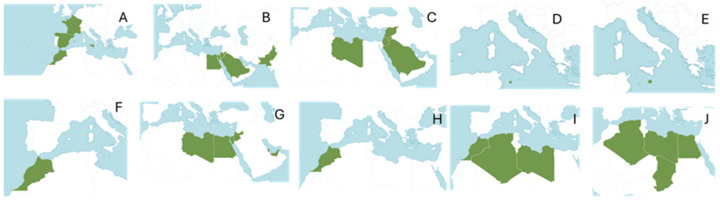
Geographical distribution of the species of the genus *Chiliadenus* (marked in green) (**A**) *C. glutinosus*, (**B**) *C. montanus*, (**C**) *C. iphionoides*, (**D**) *C. lopadusanus*, (**E**) *C. bocconei*, (**F**) *C. antiatlanticus*, (**G**) *C. candicans*, (**H**) *C. hesperius*, (**I**) *C. rupestris*, (**J**) *C. sericeus*) [https://powo.science.kew.org/taxon/urn:lsid:ipni.org:names:1016435-1, accessed on 28 August 2024].

**Figure 2 plants-14-00205-f002:**
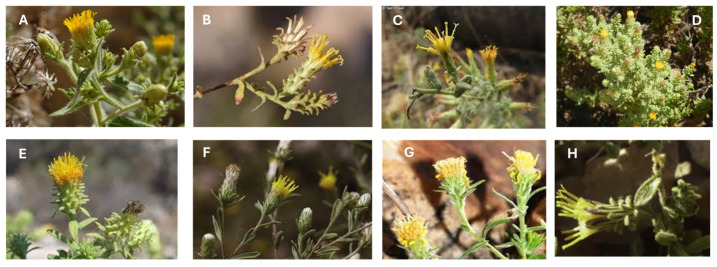
Images of aerial parts of some species of the genus *Chiliadenus* (**A**) *C. glutinosus*, (**B**) *C. montanus*, (**C**) *C. iphionoides*, (**D**) *C. lopadusanus*, (**E**) *C. bocconei*, (**F**) *C. antiatlanticus*, (**G**) *C. hesperius*, (**H**) *C. rupestris*).

**Figure 3 plants-14-00205-f003:**
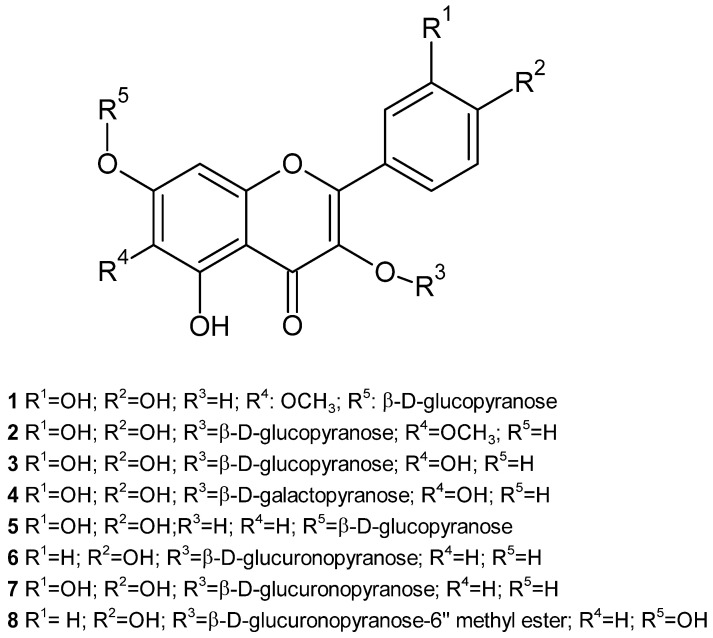
Chemical structure of the flavonoids isolated from *J. glutinosa* (*C. glutinosus*) (**1**–**8**).

**Figure 4 plants-14-00205-f004:**
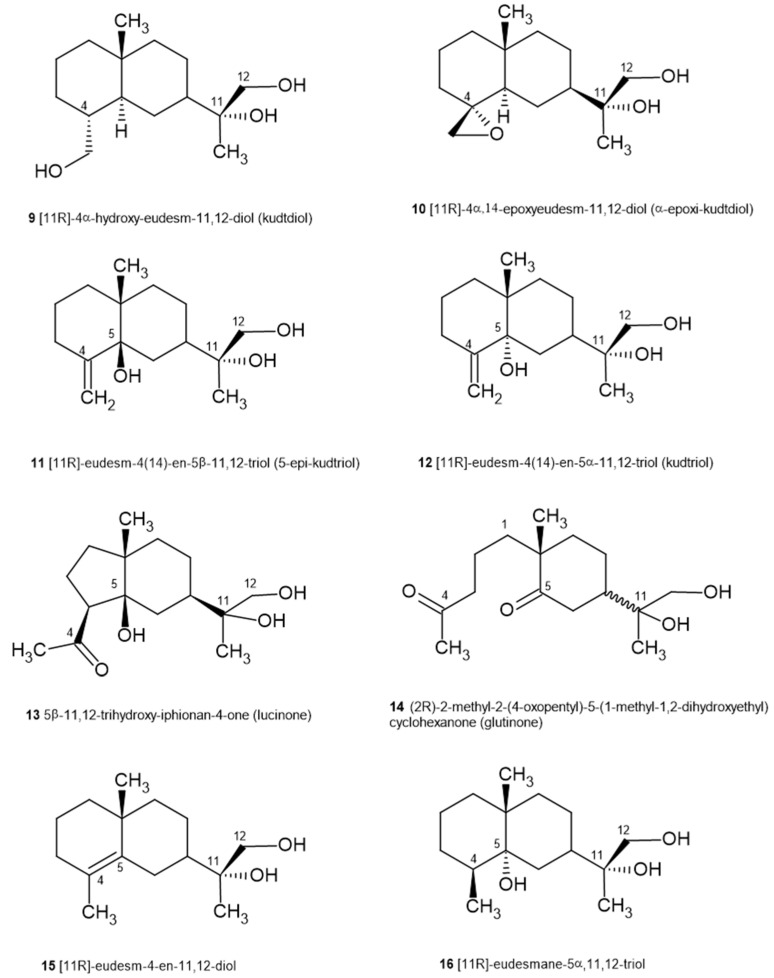
Chemical structure of the sesquiterpenes isolated from *J. glutinosa* (*C. glutinosus*) (**9**–**16**).

**Figure 5 plants-14-00205-f005:**
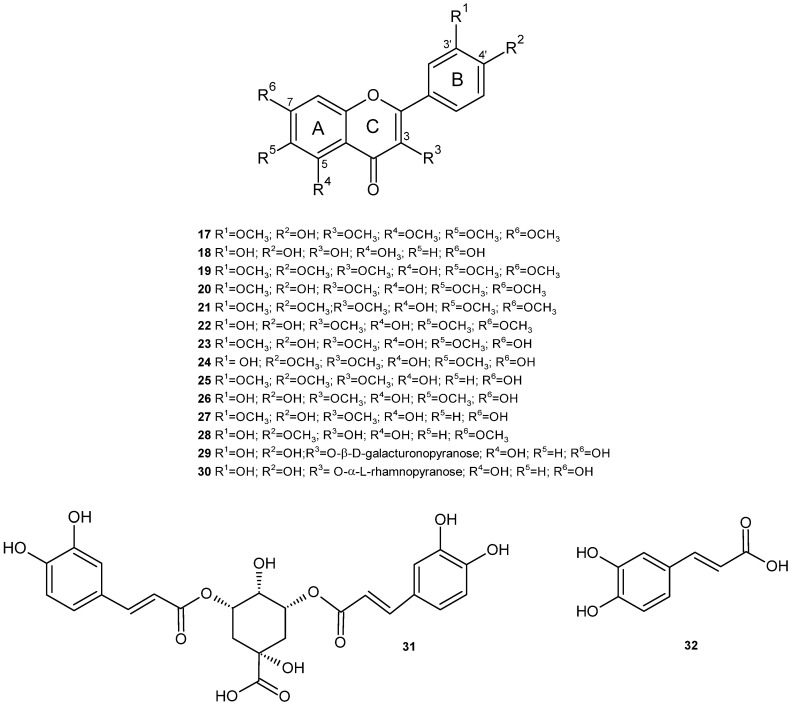
Chemical structure of the phenolic compounds isolated from *J. montana (C. montanus)* (**17**–**32**).

**Figure 6 plants-14-00205-f006:**
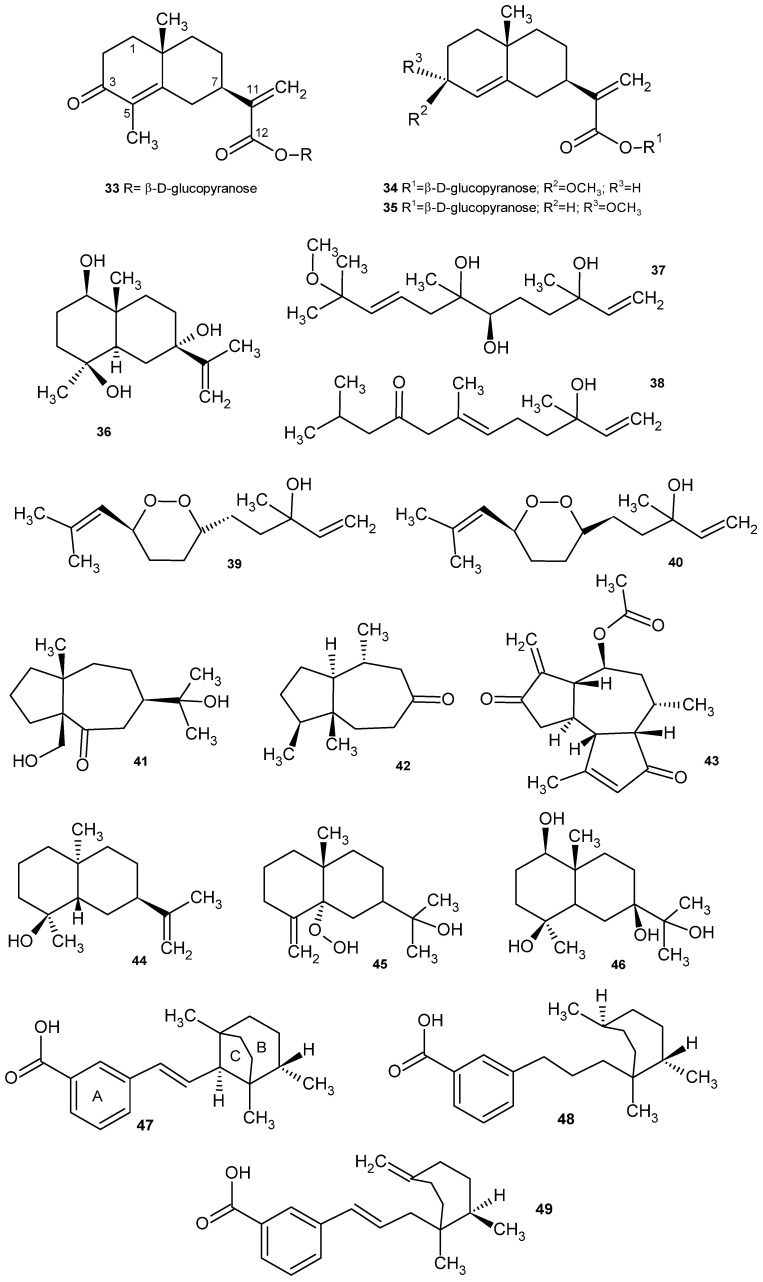
Chemical structure of the terpenes isolated from *J. montana (C. montanus)* (**33**–**49**).

**Figure 7 plants-14-00205-f007:**
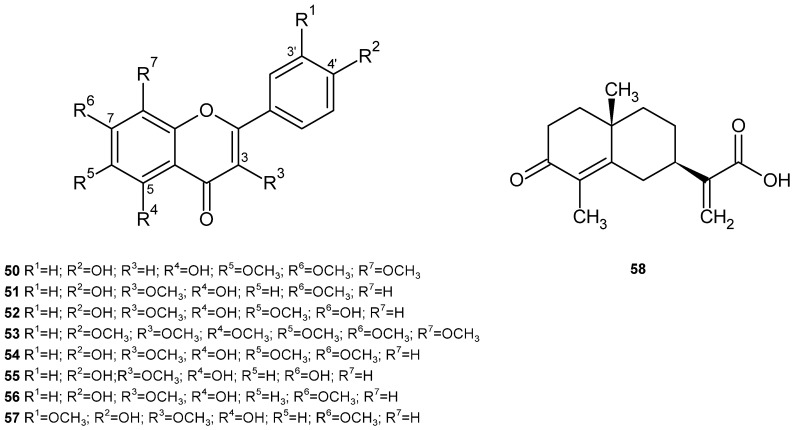
Chemical structure of the compounds isolated from *V. iphionoides* (*C. iphionoides*) (**50–58**).

**Figure 8 plants-14-00205-f008:**
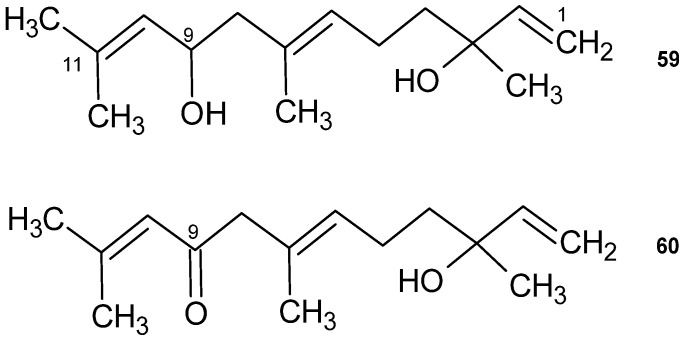
*Chemical structure* of non-cyclic sesquiterpenes isolated from *C. lopadusanus* (**59–60**).

**Figure 9 plants-14-00205-f009:**
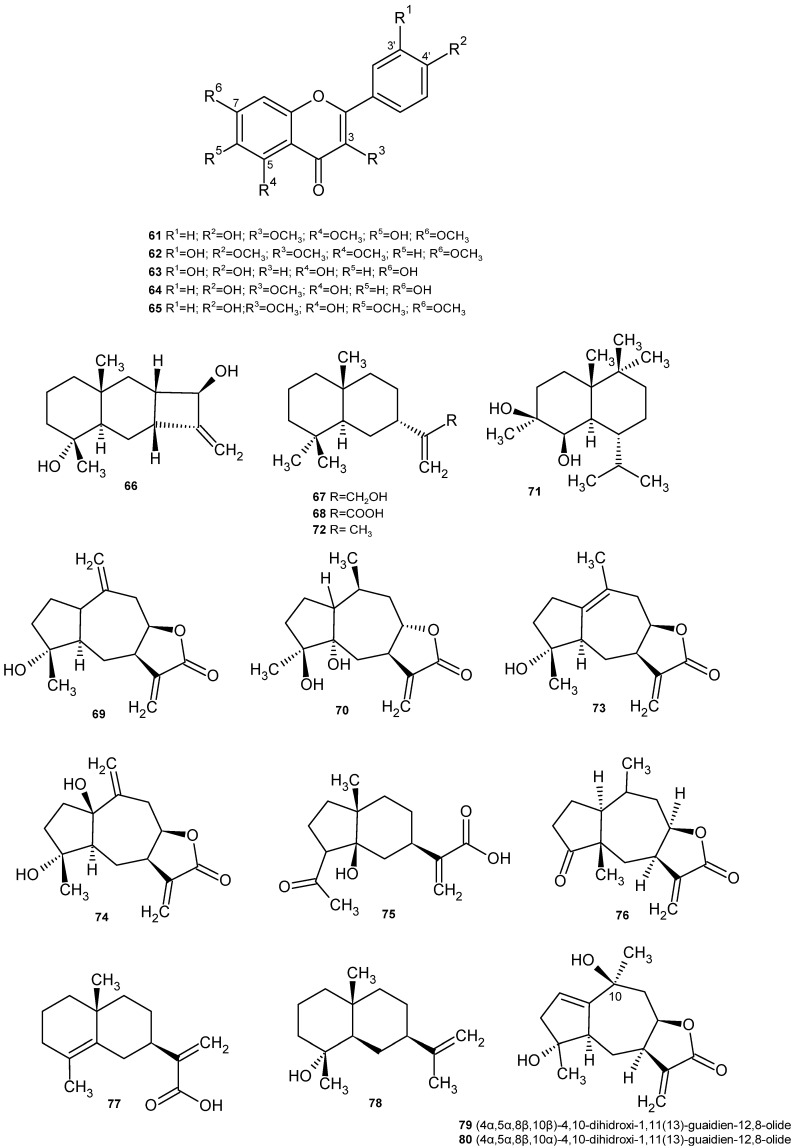
Chemical structure of the compounds isolated from *C. candicans* (**61–80**).

**Table 1 plants-14-00205-t001:** Species belonging to the genus *Chiliadenus* Cass. and their synonyms.

Plant Species	Synonymies
*Chiliadenus glutinosus* (L.) Fourr.	*Jasonia glutinosa* (L.) DC *Jasonia saxatilis* (Lam) Guss *Chiliadenus saxatilis* (Lam.) Brullo *Erigeron glutinosus* L. *Inula saxatilis* Lam
*Chiliadenus montanus* (Vahl) Brullo	*Jasonia montana* (Vahl) Botsch. *Varthemia montana* (Vahl.) Boiss. *Chrysocoma montana* (Vahl.) Symb. *Inula conyzoides* DC. *Linosyris montana* DC. *Varthemia conyzoides* Boiss.
*Chiliadenus iphionoides* (Boiss. and Blanche) Brullo	*Blanchea iphionioides* (Boiss. and C.I. Blanche) Boiss. *Iphiona iphionoides* (Boiss. and C.I. Blanche) Benth. and Hook.f. ex B.D. Jacks. *Jasonia iphionoides* (Boiss. and C.I. Blanche) Botsch. *Varthemia iphionoides* Boiss. and C.I. Blanche
*Chiliadenus lopadusanus* Brullo	*Jasonia lopadusana* (Brullo) M. Pardo and R. Morales
*Chiliadenus bocconei* Brullo	*Jasonia bocconei* (Brullo) M. Pardo and R. Morales
*Chiliadenus antiatlanticus* (Emb. and Maire) F. Gómez	*Jasonia antiatlántica* (Emb. and Maire) Gómiz *Jasonia antiatlantica* (Emb. and Maire) Gómiz *Jasonia glutinosa* var. *antiatlantica* Emb. and Maire
*Chiliadenus candicans* (Delile) Brullo	*Chrysocoma candicans* Delile *Jasonia candicans* (Delile) Botsch. *Linosyris candicans* (Delile) DC. *Varthemia candicans* (Delile) Boiss.
*Chiliadenus hesperius* (Maire and Wilczek) Brullo	*Jasonia hesperia* Maire and Wilczek
*Chiliadenus rupestris* (Pomel) Brullo	*Jasonia rupestris* Pomel
*Chiliadenus sericeus* (Batt. and Trab.) Brullo	*Jasonia sericea* Batt. and Trab. *Varthemia sericea* (Batt. and Trab.) Diels

**Table 3 plants-14-00205-t003:** Main chemical compounds identified in *J. montana* (*C. montanus*).

Compound	Reference
Flavonoid aglycons	
Quercetagetin 3,5,6,7,3′-pentamethyl ether (**17**)	[55]
Quercetin (**18**)	[54]
3, 6, 7, 3′, 4′-pentamethoxy quercetin (artemetin) (**19**)	[54]
3, 6, 7, 3′-tetramethoxy quercetin (chrysosplenetin) (**20**)	[54]
3, 6, 3′, 4′-tetramethoxy quercetin (bonanzin) (**21**)	[54]
3, 6, 7-trimethoxy quercetin (chrysosplenol-D) (**22**)	[54]
3, 6, 3′-trimethoxy quercetin (jaceidin) (**23**)	[54]
3, 6, 4′-trimethoxy quercetin (centaureidin) (**24**)	[56]
3, 3′, 4′-trimethoxy quercetin (**25**)	[54]
3, 6-dimethoxy quercetin (**26**)	[54]
3, 3′-dimethoxy quercetin (**27**)	[54]
7, 4′-dimethoxy quercetin (**28**)	[54]
Flavonoid glycosides	
Quercetin-3-O-β-D-galacturonopyranoside (**29**)	[54]
Quercetin-3-O-β-D-glucopyranoside (**3**)	[54]
Patuletin-7-O-β-D-glucopyranoside (**1**)	[54]
Quercetin-3-O-α-L-rhamnopyranoside (quercitrin) (**30**)	[54]
Quercetin-3-O-β-D-glucuronopyranoside (**7**)	[54]
Phenolic acids	
3,5-dicaffeoyl-quinic acid (**31**)	[54]
Caffeic acid (**32**)	[54]
Sesquiterpenes	
3-oxo-γ-costic acid β-D-glucopyranoside ester (**33**)	[57]
3β-methoxy isocostic acid (**34**)	[57]
3α-methoxy isocostic acid (**35**)	[57]
Eudesm-11,13-ene-1β,4β,7α-triol (**36**)	[57]
3,6,7-trihydroxy-11-methoxy-3,7,11-trimethyldodeca-1,9-diene (chiliadenol A) (**37**)	[57]
3-hydroxy-3,7,11-trimethyl-1,6-dodecadien-9-one (chiliadenol B) (**38**)	[57]
3-hydroxy-3,11-dimethyl-6β,9α-epidioxy-dodeca-1,7(14),10-triene (chiliadenol C) (**39**)	[57]
3-hydroxy-3,11-dimethyl-6α,9α-epidioxy-dodeca-1,10,7(14)-ene (chiliadenol D) (**40**)	[57]
(3aR*, 6R*, 8aR)-3a-(hydroxymethyl)-6-(2-hydroxy-propan-2-yl)-8a methyl-octahydrazulen-4(5H)-one (jasomontanone) (**41**)	[58]
4-hydroxy-5,10-dimethyl-octahydro-azulen-8-one (jasonone) (**42**)	[59]
Montanone (**43**)	[60]
Intermedeol (**44**)	[61]
5α-hydroperoxy-β-eudesmol (**45**)	[61]
1β,4β,7β,11-tetrahydroxyeudesmane (**46**)	[61]
Diterpenes	
(1 E)-2-((2 S)-1,2,5-trimethylbicyclo[3.2.1]octan-8-yl)vinyl]benzene-3-carboxylic acid (jasonin a) (**47**)	[56]
3-(2S,5S)-1,2,5-trimethylcycloheptanyl)propyl]benzene-3-carboxylic acid (jasonin b) (**48**)	[56]
(1 E)-3-((7 R)-1,7-dimethyl-4-methylenecycloheptanyl)prop-1-enyl]benzene-3-carboxylic acid (jasonin c) (**49**)	[56]

**Table 5 plants-14-00205-t005:** Main compounds isolated from *C. candicans*.

Compound	Reference
Flavonoids	
6,4′-dihydroxy-3,5,7-trimethoxyflavone (6-hydroxykaempferol 3,5,7-trimethyl ether) (**61**)	[122]
3′-hydroxy-3,5,7,4′-tetramethoxyflavone (quercetagetin 3,5,7,4′-tetramethyl ether) (**62**)	[122]
Luteolin (**63**)	[122]
Isokaempferide (**64**)	[122]
Penduletin (**65**)	[122]
Jaceidin (**23**)	[122]
Terpenes	
Jasonol (**66**)	[123]
12-methylether) (**67**)	[123]
7-epi-ilicic acid (**68**)	[123]
lβ-hydroxy-8-epi-inuviscolide (**69**)	[123]
5α-hydroxy-10,14-dihydroinuviscolide (**70**)	[123]
Cadinane-triol (**71**)	[123]
Isointermedeol (**72**)	[123]
Pseudoivalin (**73**)	[123]
8-epi-inuviscolide (**74**)	[123]
5β-hydroxy-4-oxo-ll(13)-dehydroiphionan-12-oic acid (**75**)	[123]
Confertin (**76**)	[122]
4,11(13)-eudesmadien-12-oic acid (**77**)	[122]
11-eudesmen-4-ol (**78**)	[122]
(4α,5α,8β,10β)-4,10-dihidroxi-1,11(13)-guaidien-12,8-olide (**79**)	[122]
(4α, 5α, 8β,10α)-4,10-dihidroxi-1,11 (13)-guaidien-12,8-olide (**80**)	[122]

**Table 6 plants-14-00205-t006:** Main pharmacological activities described for the different species (crude extracts and isolated compounds) of the *Chiliadenus* genus.

Plant Species	Pharmacological Activity	Reference
*Chiliadenus glutinosus*	Crude extracts - Antiinflammatory - Antioxydant - Neuroprotective - Cytotoxic - Antispasmodic - Antihypertensive - Antiprotozoal - In metabolic disorders Isolated compounds - Antiprotozoal - Antiinflammatory	[32,33] [32,33,38,39] [34] [38] [40] [41] [43,44,45] [46] [47] [48]
*Chiliadenus montanus*	Crude extracts - Antioxidant - Antiinflammatory - Antihepatotoxic - Antimicrobial - Antiobesity - Antidiabetic - Cytotoxic - Anticholinesterase - Antiplatelet Isolated compounds - Antioxidant - Cytotoxic - Antiinflammatory - Phytotoxic	[67,68,69,71,72,73,76,80] [59] [70,74,75,76] [56,65,81,82] [71] [62,76,77,78,79] [83] [84] [85] [53,61,86,87] [53,54,61,64,88] [63] [90]
*Chiliadenus iphionoides*	Crude extracts - Antimicrobial - Antioxidant - Antidiabetic - Cytotoxic Isolated compounds - Antispasmodic - Antimicrobial - Antiplatelet - Antioxidant	[95,102,103] [96,103] [104,105,106,107] [96,108,109] [110] [95,97] [111] [96]
*Chiliadenus lopdusanus*	Crude extracts - Antimicrobial	[112]
*Chiliadenus antiatlanticus*	Crude extracts - Antimicrobial Isolated compounds - Antimicrobial - Cytotoxic	[121] [113] [113]
*Chiliadenus candicans*	Crude extracts - Antimicrobial - Anti-amyloidogenic - Cytotoxic	[65] [84] [125]

## Data Availability

Not applicable.

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
