# Peer review of "The Genus *Chiliadenus*: A Comprehensive Review of Taxonomic Aspects, Traditional Uses, Phytochemistry and Pharmacological Activities"

_plants, 2025, doi:10.3390/plants14020205_

Round 1

Reviewer 1 Report

Comments and Suggestions for Authors

The review manuscript “The genus Chiliadenus: A comprehensive review of taxonomic aspects, traditional uses, phytochemistry and pharmacological activities” describes the taxonomic diversity of the genus Chiliadenus as well as the traditional uses of plants from this genus. Additionally, it establishes a link between traditional uses and the biological activities associated with plants of the genus, and further highlights this relationship with the chemical structures of the molecules present. Overall, the manuscript is well written and organized, effectively presenting the objective and providing the reader with the necessary and comprehensive information on the subject matter. However, some minor modifications are required to achieve the quality necessary for acceptance for publication. Among these are the addition of citations in the introduction to better support it, and improvements in the formatting of certain words in the text. Upon addressing the requests presented in the comments, I consider the manuscript suitable for publication.

Comments to the authors:

Comment 1. Please consider replacing some keywords already included in the title with different ones. This will increase the chances of your work being found in search databases.

Comment 2. In the first paragraph of the introduction (lines 28-35), some statements are made without citations to support them. Please search for and cite articles that provide evidence for the statements presented.

 Comment 3. In the second paragraph of the introduction (lines 36-49), the authors make some statements without citing articles to substantiate them. Please add works that substantiate the statements made and include the respective citations at the end of each statement.

 Comment 4. The introduction as a whole is well written; however, it requires the citation of articles to scientifically support the statements made. Please add these citations to substantiate what has been written.

Comment 5. The methodology section should be better written, specifying more clearly the descriptors used in the search, as well as the inclusion and exclusion criteria used to gather information.

 Comment 6. Please specify the number of documents/articles found in each database, as well as the total number of documents found in the search and the final number of articles used in this review after exclusions, if possible.

 Comment 7. In some places in Table 1, the authors' names are italicized along with the species names. Modify so that only the species names are in italics.

 Comment 8. Line 209. The name of the tribe does not need to be in italics.

 Comment 9. Lines 213-215. The names of the genus and species are not italicized. Correct this.

 Comment 10. Lines 273-279. The authors make a statement but do not cite any work to support it. Please add a citation to support this statement.

 Comment 11. Lines 307 and 308. The species name needs to be in italics.

 Comment 12. Lines 735-736. The species name is not italicized. Check and correct this.

 Comment 13. Lines 1009-1010. During the writing process, the text formatting divided the paragraph into two. Correct this.

 Comment 14. Line 1019. A character was typed before the species name. Check the necessity of this character.

 Comment 15. Line 1097. In some places in the text, "et al." is not in italics, while in others it is. Standardize by italicising "et al." throughout the entire text.

 Comment 16. Line 1142. The species name is not in italics.

Comment 17. Line 1191. The species name is not in italics.

Although well written, the manuscript is not ready for publication in this journal. Some considerations must be taken into account so that the manuscript has the quality required to be accepted.

Author Response

We greatly appreciate all your comments to improve the article and apologize for any formatting errors. Below we respond to your considerations and comments.

Comment 1. Please consider replacing some keywords already included in the title with different ones. This will increase the chances of your work being found in search databases.

Response 1. Thank you for this suggestion. We have revised the keywords and replaced those already included in the title with new terms that enhance the visibility and searchability of our manuscript. We have replaced "Pharmacological activities" with "Biological activities”, "Phytochemistry" with "Bioactive compounds" and "Traditional uses" with "Folk medicine."

Comment 2. In the first paragraph of the introduction (lines 28-35), some statements are made without citations to support them. Please search for and cite articles that provide evidence for the statements presented.

Response 2. Thank you for your observation regarding the lack of citations in the first paragraph of the introduction (lines 28-35). I would like to clarify that the statements in these lines are indeed supported by Reference [1], which is cited in the subsequent paragraph. This reference corresponds to the article: Fabricant, D.S. and Farnsworth N.R. The value of plants used in traditional medicine for drug discovery. Environ Health Perspect, 2001. 

This is a widely recognized and authoritative article that thoroughly discusses the relevance of natural products in drug discovery, including their structural diversity, pharmacokinetic properties, and the correlation between traditional uses and therapeutic applications. Given the broad scope of this reference, it provides sufficient support for the general assertions presented in lines 28-35.

To address your concern, I will include a direct citation to Reference [1] at the end of line 35 to explicitly indicate that this source supports the statements made in that paragraph. The reason for this addition is to enhance clarity and ensure that the citation is immediately visible to readers.

Regarding the inclusion of additional references, I believe that the extensive coverage provided by Fabricant and Farnsworth in their review is sufficient to substantiate the key concepts mentioned. Including multiple references for these general statements could lead to unnecessary redundancy, as the cited article already encapsulates the essence of the points discussed.

I hope this explanation clarifies the approach taken in this section. However, I remain open to incorporating additional references if you feel they are strictly necessary to complement or clarify specific aspects.

Comment 3. In the second paragraph of the introduction (lines 36-49), the authors make some statements without citing articles to substantiate them. Please add works that substantiate the statements made and include the respective citations at the end of each statement.

Response 3. As mentioned in my response to your previous comment about lines 28–35, the statements included in this section are also supported by Reference [1], which is a well-recognized and comprehensive article addressing the principles discussed. The article by Fabricant and Farnsworth (The value of plants used in traditional medicine for drug discovery, Environ Health Perspect, 2001) provides a detailed overview of the structural complexity of natural products, their selective interactions with biological targets, and their favorable pharmacokinetic profiles. These are the foundational concepts upon which this section is based, and Reference [1] remains sufficient to substantiate the claims made in lines 36–49. I appreciate your suggestion and understand the importance of ensuring clarity and robust support for all statements. 

Comment 4. The introduction as a whole is well written; however, it requires the citation of articles to scientifically support the statements made. Please add these citations to substantiate what has been written.

Response 4. Thank you for your observation regarding the need for citations to substantiate the statements made in the introduction. I would like to clarify that many of the statements, particularly those in the later paragraphs of the introduction, reflect the authors' own conclusions and interpretations based on a comprehensive review of the available literature. These general impressions are essential for framing the context of our work and articulating the objectives that guided the preparation of this manuscript.

As such, these statements are not directly attributable to specific studies or authors but rather represent a synthesis of the knowledge acquired through our review. Adding excessive citations in these cases could detract from the narrative flow and the focus on presenting a coherent rationale for our study.

That said, we believe that the references already included—particularly Reference [1], which serves as a foundational source for several key points—are sufficient to support the claims made in the introduction. However, we are open to revisiting this section and considering additional citations if deemed absolutely necessary to improve clarity or support specific statements.

Comment 5. The methodology section should be better written, specifying more clearly the descriptors used in the search, as well as the inclusion and exclusion criteria used to gather information.

Response 5. We have carefully considered your suggestion to improve the clarity and specificity of the search terms, as well as the inclusion and exclusion criteria used to gather information. In response to your comment, we have revised the methodology section to provide a more detailed and transparent description of the search process. Specifically, we have:

  • Clarified the search terms and descriptors: We have listed all the primary and additional search terms used, such as Chiliadenus, Jasonia, Inula, Erigeron, as well as the relevant species names, including their known synonyms. This ensures that the search was exhaustive and covered all relevant publications related to both genera and their synonymies.
  • Expanded the inclusion and exclusion criteria: We have now specified the exact criteria used to select relevant articles, including studies focusing on taxonomic classifications, chemical composition, pharmacological activities, and traditional uses. We have also outlined the articles excluded from the review, ensuring a clear rationale for what was included.
  • Explained the database selection and search process: We have detailed the specific databases used for the search (PubMed, Web of Science, Scopus, etc.), and we have also clarified that no time limits were imposed on the search years to ensure comprehensive coverage of all available literature. This approach was intended to provide the most up-to-date and relevant information for the review.
  • Justified the use of global taxonomic databases: To ensure the accuracy of the taxonomic nomenclature and synonymies, we consulted authoritative resources such as the World Flora Online (WFO) Plant List and Plants of the World Online (POWO), which are widely recognized for providing updated and reliable information on plant species classification.

Comment 6. Please specify the number of documents/articles found in each database, as well as the total number of documents found in the search and the final number of articles used in this review after exclusions, if possible.

Response 6. We have prepared a table with the number of articles found in each database, which we are happy to share with you for your reference. However, we would like to clarify that we did not include these specific details in the manuscript as we did not consider it necessary for the purposes of this review. Given that our work is not a systematic review but rather a comprehensive bibliographic overview, we focused on summarizing the most relevant articles related to the genus Chiliadenus and its pharmacological and traditional uses, as well as their chemical composition.

We believe that the information provided in the methodology section is sufficient to give readers a clear understanding of the search strategy and the general approach used to gather the articles, including the selection criteria and the resources consulted. We feel that providing overly specific numbers might not be essential in the context of this type of review and could unnecessarily complicate the section.

That said, we are more than willing to include the requested table as supplementary material or as an appendix if you deem it necessary, and we hope that the rest of the information we have provided is adequate to substantiate the thoroughness of our search.

Bases de datos

Términos de búsqueda

Chiliadenus

Jasonia

Inula

(I. saxatilis)

Erigeron

(E. glutinosus)

PubMed

17

23

680 (0)

460 (0)

Scopus

30

56

1685 (3)

1365 (1)

Web of Science

31

64

1275 (0)

948 (0)

ScienceDirect

50

109

2309 (25)

2107 (4)

SciFinder

34

69

6215 (1947)

2861 (734)

Wiley Online Library

29

47

1365 (65)

2549 (14)

ACS Publications

1

10

202 (2)

135 (0)

SpringerLink

50

68

3727 (313)

3708 (24)

Google Scholar

704

1730

51200 (4470)

60600 (282)

Comment 7. In some places in Table 1, the authors' names are italicized along with the species names. Modify so that only the species names are in italics.

Response 7. The format of the authors' names has been corrected, removing the italics

Comment 8. Line 209. The name of the tribe does not need to be in italics.

Response 8. Italics have been removed from the tribe name

Comment 9. Lines 213-215. The names of the genus and species are not italicized. Correct this.

Response 9. Italics have been placed on the names of genus and species

Comment 10. Lines 273-279. The authors make a statement but do not cite any work to support it. Please add a citation to support this statement.

Response 10. The corresponding citation has been included

Comment 11. Lines 307 and 308. The species name needs to be in italics.

Response 11. Species names have been changed to italics

Comment 12. Lines 735-736. The species name is not italicized. Check and correct this.

Response 12. Has been modified

Comment 13. Lines 1009-1010. During the writing process, the text formatting divided the paragraph into two. Correct this.

Response 13. Has been fixed

Comment 14. Line 1019. A character was typed before the species name. Check the necessity of this character.

Response 14. The reason for placing square brackets around this taxonomic name is to indicate a synonymy. In this case, Chiliadenus lopadusanus Brullo is the currently accepted name, while Jasonia lopadusana (Brullo) M. Pardo & R. Morales is the synonym that has been historically used in some sources.

Comment 15. Line 1097. In some places in the text, "et al." is not in italics, while in others it is. Standardize by italicising "et al." throughout the entire text.

Response 15. It has been standardized by italicizing it throughout the text

Comment 16. Line 1142. The species name is not in italics.

Response 16. Has been fixed

Comentário 17. Linha 1191. O nome da espécie não está em Itálico.

Response 17. Has been fixed

We thank you again for your time and for the comments and suggestions you have provided.

Reviewer 2 Report

Comments and Suggestions for Authors

The authors provided interesting, qualitative and really comprehensive data about genus Chiliadenus and its ten species. I only have one important remark: It looks like different authors wrote different parts of the manuscript and no one carefully read the whole paper and uniform the text, style, abbreviations etc. Please do it. Also, it would be nice to include pictures of listed species.

Some of the comments:

-Family names are usually italicized

-Ln55 Artemisia annua italic

-Ln64 Chiliadenus Cass italic; Ln70 as well and on the rest of the paper

-Ln107 fullstop (.) should be deleted

-Ln112, Ln118-120 Ln124-125, Ln128-129,  Ln213-215 secies and genus names italic, or why you wrote those names non-italic?

-Please uniform the paper, in some places some names are italic on the other places they are not.

-Are only paper in English language were selected or other languages were included as well?

-In Table1 some words should not be italicized

-Also, when you first time use full species name in rest of the paper genus name can be shortened 

-Ln339-340 introduced full name for listed abbreviation, the same in future text below

-Ln351 ID typo mistake

-Ln407 please uniform the italic and non-italic words, in all text before in vitro is not italic

Ln438 LC-MS as this and other techniques are mentioned earlier and full name was not introduced, please correct it on all relevant places

-Ln520 LOX is abbreviation for what?

-Ln1000 ADP is abbreviation for what?

-Ln1074 minimum inhibitory concentration or MIC as abbreviation was introduced earlier in manuscript

-Ln1078  Staphylococcus to aureus typo mistake

-Ln1231-1232 remove hyperlink

Author Response

Thank you very much for accepting the review of our work. Below, we are pleased to respond to your comments. We have carefully reviewed the manuscript and unified the style, abbreviations, etc. Additionally, we have included a figure with images available in the literature of the main Chiliadenus species. We apologize for the formatting errors.

Comments 1: Family names are usually italicized

Response 1: We have italicized the names of the Families.

Comments 2: Ln55 Artemisia annua italic

Response 2: It has been italicized.

Comments 3: Ln64 Chiliadenus Cass italic; Ln70 as well and on the rest of the paper

Response 3: The italics have been reviewed throughout the document. We apologize once again, as it was an issue caused by changing the text format.

Comments 4: Ln107 fullstop (.) should be deleted

Response 4: The period (.) has been removed.

Comments 5: Ln112, Ln118-120 Ln124-125, Ln128-129,  Ln213-215 species and genus names italic, or why you wrote those names non-italic?

Response 5: The names of all species and genera mentioned in those lines, as well as throughout the rest of the document, have been italicized.

Comments 6: Please uniform the paper, in some places some names are italic on the other places they are not.

Response 6: The italic format has been standardized throughout the document.

Comments 7: Are only paper in English language were selected or other languages were included as well?

Response 7: We selected articles in English as well as articles in Spanish, as some plants of the genus Chiliadenus grow in Spain, and many Spanish researchers published their articles in journals edited in Spanish.

Comments 8: In Table1 some words should not be italicized

Response 8: The italics have been removed from the species identifier names.

Comments 9: Also, when you first time use full species name in rest of the paper genus name can be shortened 

Response 9: The names of all genera have been abbreviated starting from the second time they appear in the text, except in section 3. Taxonomic considerations on the genus Chiliadenus and its relationship to the genus Jasonia. The aim is to provide the reader with complete and clear information on the nomenclature and taxonomy of the species under study and their relationships with each other.

Comments 10: Ln339-340 introduced full name for listed abbreviation, the same in future text below

Response 10: The acronyms that appear for the first time in the text have been defined.

Comments 11: Ln351 ID typo mistake

Response 11: We have replaced "ID" with "1D".

Comments 12: Ln407 please uniform the italic and non-italic words, in all text before in vitro is not italic

Response 12: "In vitro" has been standardized throughout the text and italicized.

Comments 13: Ln438 LC-MS as this and other techniques are mentioned earlier and full name was not introduced, please correct it on all relevant places

Response 13: "LC" has been defined with capital letters at the beginning of each word, and the definition of "MS" has been removed, as it was defined earlier when it first appeared in the text.

Comments 14: Ln520 LOX is abbreviation for what?

Response 14: LOX is the abbreviation for lipoxygenase. It is defined in line 405.

Comments 15: Ln1000 ADP is abbreviation for what?

Response 15: Adenosine Diphosphate. The acronym has been defined in line 1000 as it is the first time it appears in the text.

Comments 16: Ln1074 minimum inhibitory concentration or MIC as abbreviation was introduced earlier in manuscript

Response 16: The acronym for minimum inhibitory concentration (MIC) has been included, as it had indeed been mentioned earlier.

Comments 17: Ln1078  Staphylococcus to aureus typo mistake

Response 17: It has been corrected.

Comments 18: Ln1231-1232 remove hyperlink

Response 18: The hyperlink has been removed.

We thank you again for your time and for the comments and suggestions you have provided.

Reviewer 3 Report

Comments and Suggestions for Authors

To my opinion, the manuscript is very well written and structured. It is obvious that the authors have invested a lot of time and efforts.

I believe that it will be good if the authors provide photos of some of the species of the genus Chiliadenus especially to the ones that look very similar to each other.

Along with that that I think that a table summarizing the pharmacological activities of the species and a figure illustrating its mechanism should stand very good in the manuscript. At that moment the phytochemistry part of the review is dominating.

Author Response

Thank you very much for taking the time to review our manuscript. We respond to your suggestions and comments below.

Comments: To my opinion, the manuscript is very well written and structured. It is obvious that the authors have invested a lot of time and efforts.

I believe that it will be good if the authors provide photos of some of the species of the genus Chiliadenus especially to the ones that look very similar to each other.

Along with that that I think that a table summarizing the pharmacological activities of the species and a figure illustrating its mechanism should stand very good in the manuscript. At that moment the phytochemistry part of the review is dominating.

Response: Thank you very much for your positive evaluation of our manuscript and for acknowledging the time and effort we have dedicated to it. We greatly appreciate your thoughtful suggestions, which have helped improve the quality of our work. In response to your comments, we have taken the following steps:

We have included photos of some of the species of the genus Chiliadenus, particularly those that are morphologically similar, to provide better visual clarification for readers.

We have also added a table (Table 2) summarizing the pharmacological activities of the crude extracts and isolated compounds reported in the literature for the different species of Chiliadenus.

Regarding the inclusion of a figure illustrating the mechanisms of action of the plants, we would like to clarify the following: 

Lack of Mechanistic Studies: For many of the pharmacological activities described in the literature, the specific mechanisms of action have not yet been elucidated. Most studies report biological effects (e.g., anti-inflammatory, antimicrobial, antioxidant activities) without providing detailed insights into the molecular pathways or targets involved.

Diversity of Pharmacological Activities: The genus Chiliadenus has been reported to exhibit a wide range of pharmacological activities, likely mediated by distinct and diverse mechanisms. Attempting to generalize these mechanisms into a single figure would risk oversimplifying the complexity of these effects and potentially misrepresenting the findings in the literature.

 Scientific Integrity: We believe it is crucial to ensure that all information presented in the manuscript is well-supported by existing evidence. Since the mechanisms of action remain largely unexplored for many activities, any figure created at this point would necessarily rely on speculation rather than data, which we feel would detract from the scientific rigor of the manuscript.

We hope this explanation provides clarity regarding the challenges of including a mechanism-focused figure in the current review. We appreciate your understanding and trust that the addition of the photos and the table adequately addresses your valuable suggestions to enhance the manuscript. Thank you again for your thoughtful feedback.

Reviewer 4 Report

Comments and Suggestions for Authors

This manuscript provides a review of the chemical composition and pharmacological activities of plants in the genus Chiliadenus. The literature data collection is comprehensive, and the manuscript explanation is reasonable. But the manuscript still needs to be revised in terms of expression.

1. In the abstract, there are too many descriptions of the background, without a conclusive summary of the results, and a lack of explanation of the results to attract readers.

2. The format in the text needs further standardization and revision, such as using italics for the genus Latin names, and abbreviations are required when the genus name appears for the second time.

3. In the results section, the manuscript derived physiological and pharmacological activities, but the description of these results in this section is not serious enoughIt is best to describe the pharmacological activity of the compound.

4. Multiple genus names appear in the manuscript, please verify and explain in detail.

5. The Conclusions and perspectives section lacks specific description of the results and is relatively scattered.

6. Please provide figures of the plants mentioned in the manuscript for readers to quickly understand.

Author Response

Thank you for your thoughtful and constructive comments on our manuscript. We greatly appreciate the time and effort you have dedicated to reviewing our work, as your suggestions have provided valuable insights for improving the clarity and quality of the article. We are now pleased to address each of your comments in detail.

Comment 1. In the abstract, there are too many descriptions of the background, without a conclusive summary of the results, and a lack of explanation of the results to attract readers.

Response 1. Thank you for your valuable feedback on the abstract. In response to the reviewer’s comments, we have expanded the abstract to include a more detailed summary of our findings while reducing the emphasis on the background information. The revised abstract now provides a clearer picture of the chemical composition, pharmacological activities, and traditional uses of Chiliadenus, as well as the identified gaps in research. We have also included a more comprehensive explanation of the potential for future research, especially regarding comparative studies and clinical trials, to better highlight the genus's therapeutic promise. We hope that this revision provides a more balanced and compelling overview of the manuscript's key results.

Comment 2. The format in the text needs further standardization and revision, such as using italics for the genus Latin names, and abbreviations are required when the genus name appears for the second time.

Response 2. We apologize for any errors in formatting the manuscript. We have addressed the points raised regarding formatting and made the necessary revisions to ensure consistency throughout the text. In particular, we have used italics for all Latin genus names according to standard conventions. In addition, I have included appropriate abbreviations for genus names when they appear a second time, as recommended.

I believe these changes are in line with your suggestions and improve the overall readability and standardization of the manuscript.

Comment 3. In the results section, the manuscript derived physiological and pharmacological activities, but the description of these results in this section is not serious enough. It is best to describe the pharmacological activity of the compound.

Response 3. Thank you for your valuable comments and suggestions. We would like to clarify that the objective of this review is to provide a general overview of the physiological and pharmacological activities associated with the Chiliadenus genus, based on previous studies, rather than delving into highly specific and detailed scientific trials or experimental results. The focus is on synthesizing the findings from various studies to present a broader understanding of the potential therapeutic properties of the plants within this genus.

Additionally, we acknowledge that, in some instances, there is a lack of detailed information regarding the specific compounds responsible for the pharmacological activities observed in the extracts. This limitation is due to the fact that many studies do not identify the precise bioactive molecules involved, or attribute the activities to a range of compounds present in the extracts.

We hope this explanation clarifies the intention behind the presentation of the results and the scope of the review. Thank you again for your insightful comments.

Comment 4. Multiple genus names appear in the manuscript, please verify and explain in detail.

Response 4. Thank you for your comment on the multiple genus names appearing in the manuscript. We would like to clarify that the genus Chiliadenus is the main focus of this review, but we also mention other genera such as Jasonia, Inula, and Erigeron due to historical taxonomic relationships. Some species within Chiliadenus were previously classified under Jasonia, and we reference these synonyms to ensure that all relevant studies, regardless of the genus name used, are considered in the review.
In addition, the genera Inula and Erigeron are mentioned due to their synonymy with certain species of Chiliadenus in some taxonomic classifications. We include studies related to these genera as they provide valuable information on the biological activities and chemical composition of the species in question.
We believe that understanding these taxonomic relationships is facilitated by the content of Table 1, which details the corresponding synonyms for each species discussed in the manuscript.
We hope that this explanation addresses your concerns and we appreciate your careful review and helpful comments.

Comment 5. The Conclusions and perspectives section lacks specific description of the results and is relatively scattered.

Response 5. In response to the suggestions regarding the "Conclusions and Perspectives" section, we have revised it to be more specific and focused. We have added more concrete details about the bioactive compounds found in Chiliadenus species and their associated pharmacological activities, such as anti-inflammatory, antioxidant, antimicrobial, and antitumor properties. Additionally, we have reorganized the section to clearly outline key areas for future research, including the need for in vivo studies, clinical trials, and taxonomic clarification of the genus. We have also emphasized the importance of an integrative approach, combining different scientific disciplines to fully explore the therapeutic potential of these plants.

Comment 6. Please provide figures of the plants mentioned in the manuscript for readers to quickly understand.

Response 6. We have included Figure 2, which shows images of some of the species belonging to the genus Chiliadenus, to make it easier for the reader to understand the possible differences.

We once again appreciate your comments and suggestions for improving the quality of the article.

Reviewer 5 Report

Comments and Suggestions for Authors

This review describes in depth the state of the art with respect to plants belonging to the genus Chiliadenus. The manuscript is well organized and only needs a few illustrations and photographs of the plants reviewed. It is helpful that traditional uses, studies in extracts and isolated compounds are presented separately. The diversity of compounds founded in this plant resource and the wide applications that they could have in therapeutics is striking. The authors very skillfully highlight the lack of preclinical and clinical pharmacological studies, as well as the biotechnological and agronomic management of certain species that look very attractive. The review has great value in this aspect because it emphasizes what remains to be studied, also recognizing limitations inherent to the complexity of the species of the genus and their classification.

Specific comments:

Kindly, italicize botanic names in the text of section 2. The same for legend in Figure 2. There are many other isolated paragraphs denoting the same problem.

Section 4. If possible, use different colors for geographical locations depicted in Figure 1.

It would be very nice if the authors could include some photographs or representative figures of the anatomy of the main plants presented in this review.

Some chemicals like CH3OH/H2O should be corrected. For instance, in L334.

L341-346. Sentence not clear.

L361: Choose one notation, GC-MS or GC/MS

L364. Replace the word element by volatile compounds.

Author Response

We greatly appreciate the time you have taken to review our manuscript and the suggestions and comments you have provided. Below is a detailed explanation of how we have addressed your suggestions.

Comment 1. Kindly, italicize botanic names in the text of section 2. The same for legend in Figure 2. There are many other isolated paragraphs denoting the same problem.

Response 1. We have reviewed the entire text to correct formatting errors, especially those related to italics in botanical names.

Comment 2. Section 4. If possible, use different colors for geographical locations depicted in Figure 1.

Response 2. Thank you for your suggestion regarding the use of different colors for the geographical locations in Figure 1. We have carefully considered your comment; However, since the figure contains separate maps for each species, we believe that using the same color throughout enhances the visual clarity and aesthetic cohesion of the image. Given that each map represents the distribution of a distinct species, there is no risk of confusion between them. We feel that maintaining a consistent color scheme helps avoid visual clutter and provides a clearer representation of the data. 

We appreciate your input and hope this explanation justifies our decision.

Comment 3. It would be very nice if the authors could include some photographs or representative figures of the anatomy of the main plants presented in this review.

Response 3. We have included Figure 2 with the available images of the main species of the genus Chiliadenus to facilitate the reader's understanding.

Comment 4. Some chemicals like CH3OH/H2O should be corrected. For instance, in L334.

Response 4. The entire document has been revised to ensure that chemical names are correctly stated.

Comment 5. L341-346. Sentence not clear.

Response 5. In response to your comment regarding the clarity of the paragraph, we have revised it to improve its readability and ensure that the information is presented more clearly. Specifically, we restructured the sentences to make the key points more straightforward and easier to follow. We believe this revision enhances the explanation of the enantioselective synthesis of lucinone by Chiu et al. and the synthesis of glutinone and its epimers by Zhang et al., while maintaining the technical details necessary for the understanding of the original research.

Comment 6. L361: Choose one notation, GC-MS or GC/MS

Response 6. We have unified the expression throughout the manuscript, such as GC-MS

Comment 7. L364. Replace the word element by volatile compounds.

Response 7. Following your suggestion, we have replaced the term "elements" with "volatile compounds"

We thank you again for your time and for the comments and suggestions you have provided.